# Radiometric dating of Middle Pleistocene carbonates: assessing consistency and performance of the U–Th and U–Pb dating methods

Timothy J. Pollard[1], Jon D. Woodhead[1], Russell N. Drysdale[1], R. Lawrence Edwards[2], Xianglei Li[3], Ashlea N. Wainwright[1], Mathieu Pythoud[2], Hai Cheng[4,5], John C. Hellstrom[1], Ilaria Isola[6], Eleonora Regattieri[7], Giovanni Zanchetta[8], and Dylan S. Parmenter[2]

[1]School of Geography, Earth and Atmospheric Sciences, University of Melbourne, Parkville, Victoria, Australia
[2]Department of Earth and Environmental Sciences, University of Minnesota, Minneapolis, MN, USA
[3]Institute of Vertebrate Paleontology and Paleoanthropology, Chinese Academy of Science, Beijing, China
[4]Institute of Global Environmental Change, Xi'an Jiaotong University, Xi'an, Shaanxi, China
[5]State Key Laboratory of Loess and Quaternary Geology, Institute of Earth Environment, Chinese Academy of Sciences, Xi'an, Shaanxi, China
[6]Istituto di Geoscienze e Georisorse, IGG-CNR, Pisa, Italy
[7]Istituto Nazionale di Geofisica e Vulcanologia INGV, Pisa, Italy
[8]Department of Earth Sciences, University of Pisa, Pisa, Italy

**Correspondence:** Timothy J. Pollard (timothy.pollard@student.unimelb.edu.au)

**Abstract.**

The U–Th and U–Pb dating methods are widely employed for radiometric dating of Pleistocene carbonates, such as speleothems and corals. The U–Th dating method has been progressively refined over recent decades, largely through advances in mass spectrometry, and is now capable of providing accurate and precise ages for carbonates as old as ~650 thousand years (ka) under

5    optimal conditions. Similarly, the U–Pb method, traditionally restricted to pre-Quaternary materials, has been adapted in recent years for dating young carbonates. As a result, there is now substantial overlap in the applicable age range of these two dating methods; however, as yet, their consistency and relative performance when dating samples in this shared interval have not been rigorously assessed.

In this study, we conduct a systematic comparison of the U–Th and U–Pb dating methods, focusing on a significant part of their

10   overlapping age range (approximately 630–430 ka). We achieve this by dating speleothem samples (secondary cave mineral deposits) from Corchia Cave, central Italy, via both techniques and evaluate their consistency and performance in terms of age precision and other key factors. Our analytical approach employs state-of-the-art multi-collector inductively coupled plasma mass spectrometry (MC-ICPMS) measurement protocols, including a U–Th measurement protocol that utilises a Faraday cup equipped with a $10^{13}$ $\Omega$ resistor to detect the low-abundance $^{234}U^+$ and $^{230}Th^+$ ion beams. This approach is particularly

15   effective for dating samples approaching the limits of the U–Th method, but also enables accurate determination of $^{238}U/^{235}U$

ratios in these samples. As a secondary objective, we compare these $^{238}U/^{235}U$ measurements with previously published speleothem values.

Our results demonstrate excellent agreement between the U–Th and U–Pb dating methods and suggest that both are capable of producing accurate and precise ages over this interval. We find that U–Pb age uncertainties are generally less predictable than U–Th age uncertainties, but, on average, do not increase significantly over the interval considered. U–Th age uncertainties, on the other hand, tend to increase in a more predictable (approximately exponential) manner. Additionally, U–Pb age uncertainties are highly dependent on the availability of sub-samples with a wide spread in parent/daughter ratios and/or highly 'radiogenic' (i.e. very low inherited-Pb) material. In our dataset, U–Pb isochron age precision surpasses that of U–Th precision at ∼520 ka, although the exact crossover point is expected to vary for different sample types and depositional settings. Overall, these findings support the prospect of obtaining accurate and internally consistent U-series chronologies spanning the Middle Pleistocene. They also suggest that, for some carbonate samples, the U–Pb dating method may provide superior age precision to the U–Th method somewhat prior to the latter reaching its upper age limit.

Finally, our results show that most (though not all) speleothems exhibit $^{238}U/^{235}U$ ratios consistent with global carbonate values, typically deviating from the conventional value of 137.88 widely adopted in geochronology, in agreement with previous studies.

## 1 Introduction

U-series dating of carbonates has provided some of the most accurate and precise age constraints on Earth processes unfolding over the Pleistocene. The U–Th disequilibrium dating technique (hereafter, U–Th method) is by far the most widely used approach for radiometric dating of Pleistocene carbonates. Its application to corals, beginning in the 1960s, and later speleothems (secondary cave mineral deposits), has led to significant advances in our understanding of Pleistocene climate. For example, U–Th dating of coral reef terraces has provided some of the most compelling evidence linking orbital insolation variations to glacial-interglacial cycles (e.g. Edwards et al., 2003), while U–Th dating of speleothems has yielded some of the most accurate and precise constraints on the timing of major climate events over the Pleistocene, including glacial terminations (Cheng et al., 2016) and millennial-scale climate oscillations associated with Dansgaard-Oeschger events (Corrick et al., 2020). U–Th dating of carbonates has also been used to address important questions in fields such as archaeology and human evolution (Hellstrom and Pickering, 2015; Pike et al., 2017), and provides data that underpins calibration of the radiocarbon age scale (Reimer et al., 2020).

U–Th analytical methods have been continuously refined over the past few decades. The development of thermal ionization mass spectrometry (TIMS) protocols in the late 1980s (Chen et al., 1986; Edwards et al., 1987), and later MC-ICP-MS protocols, led to major improvements in U–Th age precision and sample throughput (Goldstein and Stirling, 2003). Following these major breakthroughs, U–Th analytical protocols have undergone more incremental refinement over the past two decades or so.

An example of one such refinement is the advent of all-Faraday cup analytical protocols that collect the low abundance $^{234}$U$^+$ and $^{230}$Th$^+$ ion beams in a Faraday cup instead of using an ion counter (Andersen et al., 2004; Potter et al., 2005; Cheng et al., 2013). This approach allows much larger ion beams to be measured and circumvents issues associated with ion counter dead time correction and gain calibration, leading to significant improvement in age precision and an extension of the useful range of the dating method to ~650 ka in ideal circumstances (Cheng et al., 2016).

The U–Pb dating method, which has long been used for radiometric dating of igneous materials, such as zircon, has been recently adapted for dating Pleistocene carbonates, and offers a valuable means of extending speleothem and coral chronologies beyond the current ~650-ka limit of the U–Th technique (Richards et al., 1998; Woodhead et al., 2006; Denniston et al., 2008; Klaus et al., 2017). To date, the U–Pb method has been less widely applied to Pleistocene carbonates than U–Th dating. This is partly because it is more labour intensive (especially when adopting high-precision isotope dilution TIMS or MC-ICPMS approaches), but also because its application to Pleistocene carbonates is limited to samples with low inherited Pb content (Woodhead et al., 2012). Nevertheless, U–Pb dating is of similar utility, having, for example, provided radiometric age constraints on the timing of Pleistocene glacial terminations (Bajo et al., 2020), the age of key hominin fossils (Walker et al., 2006; Pickering et al., 2011), and chronologies for sub-arctic permafrost thawing over the past ~1.5 Ma (Vaks et al., 2020). Although the U–Pb method is nominally better suited to dating older material, for which sufficient time has passed for significant accumulation of radiogenic Pb, it is also well suited to dating Middle Pleistocene carbonates that typically have $^{234}$U/$^{238}$U activity ratios (hereafter, $\left[^{234}\text{U}/^{238}\text{U}\right]$, where square brackets denote an activity ratio) that are analytically resolvable from radioactive equilibrium, thus facilitating a precise and accurate disequilibrium correction (Woodhead et al., 2006). In ideal circumstances the carbonate U–Pb dating method can also provide precise ages for materials as young as ~270 ka (Cliff et al., 2010).

With recent advances in the carbonate U–Th and U–Pb dating methods, there is now considerable overlap in the applicable range of the two chronometers, especially over the interval from ~650 to ~400 ka. This overlap provides a valuable opportunity to assess consistency between the two methods, and thus, validate, to a certain extent, the assumptions underlying their use. However, since both dating approaches are based on the $^{238}$U decay series, they are not entirely independent, and for this reason, a comparison between them may not be considered to be as definitive as one involving completely independent chronometers (e.g. Min et al., 2000), all else being equal. Nevertheless, the carbonate U–Th and U–Pb dating methods do rely on different assumptions and require analysis of distinct isotopic ratios. For instance, the U–Th method relies on precise analysis of $\left[^{230}\text{Th}/^{238}\text{U}\right]$ and is highly sensitive to any inaccuracies in this measured activity ratio. In contrast, the U–Pb method requires analysis of Pb-based isotope ratios and often incorporates information from the $^{235}$U decay series as well (although most of the age resolving power lies in the $^{238}$U series for young samples). The U–Pb dating method also relies on assumptions about the initial state of intermediate products beyond $^{230}$Th in the $^{238}$U decay series, such as $^{226}$Ra and $^{231}$Pa, and closed-system behaviour with respect to these nuclides (Richards et al., 1998), which do not apply to the U–Th method.

Other independent methods of dating carbonates within this age range are available, such as electron spin resonance (ESR) dating (Radtke et al., 1988) and (U,Th)-He dating (Bender et al., 1973; Makhubela and Kramers, 2022). However, these are less well-established and tend to provide lower age precision than the U–Th and U–Pb methods, limiting their utility in assessing the accuracy of these more precise U-series-based dating approaches.

The availability of two U-series approaches to dating carbonates over the ∼650–400 ka interval also raises the question of which is most suitable to a given situation. While, U–Th dating is often assumed to be the most accurate and precise approach throughout most, if not all, of its applicable range, the U–Pb method may perform equally well or even better for samples approaching the upper age limit of U-Th dating. Consequently, selecting an optimal dating protocol for carbonates in this age range requires careful consideration. In practical terms, U–Th dating may be considered advantageous because it typically involves analysing single samples extracted from discrete positions within a given speleothem or coral sample, making it less labour intensive on a per-age basis. U–Pb dating, on the other hand, typically involves isochron approaches, whereby multiple sub-samples from within an individual growth domain are analysed to compute a single age. Although more labour intensive, isochron methods also provide a first-order check on the assumption of closed-system behaviour and more accurately account for any initial (non-radiogenic) quantity of the daughter isotopes.

The relative precision of each dating method is not yet well characterised for samples approaching their respective age limits (the upper limit for U–Th and the lower limit for U–Pb). For 'well-behaved' samples—those maintaining closed system conditions with reasonably high U and low initial Th/Pb content—the precision of U–Pb dating is likely to surpass that of U–Th somewhere before the latter reaches its nominal upper age limit. However, the exact crossover point is difficult to predict a priori because U–Pb isochron age precision is not a simple function of isotope ratio measurement precision, but also depends on various sample characteristics (e.g. Woodhead et al., 2012; Engel and Pickering, 2022). Direct comparison of samples dated using both methods therefore offers the most appropriate means of evaluating the relative performance of these two dating methods in terms of age precision and other practical considerations.

Previous studies have compared carbonate samples dated by the U–Th and U–Pb methods, although these have tended to include a relatively small number of samples across a restricted age range (e.g. Richards et al., 1998; Pickering et al., 2010). The most comprehensive comparison to date is that of Cliff et al. (2010) who obtained multiple U–Th and U–Pb ages from a flowstone that grew between ∼350–265 ka. They found good overall agreement for the youngest growth segment, although age agreement for the middle and older growth segments was more equivocal. While this study was more comprehensive than previous ones, it also had some limitations in that only one full isochron age was included in the comparison and there was some uncertainty in co-registering the U–Th and U–Pb ages to a common depth scale, precluding a more statistically rigorous assessment.

This study presents a systematic comparison of U–Th and U–Pb dating methods focusing on the older part of their overlapping age range (∼630–430 ka) where routine application of the carbonate U–Pb dating method is a more realistic prospect. We

achieve this by dating carbonate speleothem samples extracted from identical growth layers using both the U–Th and U–Pb methods, then assess consistency between them and evaluate their relative merits in terms of age precision and other practical considerations. In contrast to previous comparison studies, we compute full Tera-Wasserburg (Tera and Wasserburg, 1972) isochron ages for all samples and adopt state-of-the-art MC-ICPMS based analytical protocols. This includes use of an all-Faraday-cup U–Th measurement protocol that employs a $^{233}$U-$^{236}$U double spike for mass bias correction, and collects the low abundance $^{234}$U and $^{230}$Th isotopes in a Faraday cup fitted with a $10^{13}$ $\Omega$ resistor, which provides superior signal-to-noise ratios for small ion beams relative to more conventional $10^{11}$ $\Omega$ resistors (Pythoud, 2022). Use of this analytical protocol allows age consistency to be assessed more rigorously than in previous studies and facilitates an assessment of age precision using the most advanced U–Th measurement protocol currently available for samples in this age range.

An added benefit of adopting this high-precision U–Th analytical protocol is that it yields $^{238}$U/$^{235}$U data of sufficient accuracy and precision for assessing natural variability in this ratio. The $^{238}$U/$^{235}$U ratio is often used in U–Th and U–Pb data reduction, as well as U–Pb and Pb/Pb age calculation, and has traditionally been treated as a constant in geochronology (Hiess et al., 2012; Tissot and Dauphas, 2015). However, the development of high precision $^{238}$U/$^{235}$U measurement protocols over the past two decades has revealed variations in natural $^{238}$U/$^{235}$U ratios (typically expressed as $\delta^{238}$U values[1]) of up to a few ‰ (Weyer et al., 2008; Tissot and Dauphas, 2015). Despite these findings, only a small number of studies have examined speleothem $\delta^{238}$U. We therefore integrate our high-precision $\delta^{238}$U measurements with pre-existing speleothem data to evaluate speleothem $\delta^{238}$U variability more broadly and examine potential controlling factors.

## 2   Study site and samples

The speleothem samples analysed in this study were collected from Corchia Cave, located in the Alpi Apuane massif of Central Italy (43°59'N, 10°13'E). This cave system, which is developed in Mesozoic marbles, dolomitic marbles and dolomites, is one of the longest (∼60 km) and deepest (∼1250 m) in Europe and has been the subject of extensive past cave exploration (Piccini et al., 2008) and scientific study (Bajo et al., 2017; Drysdale et al., 2019; Isola et al., 2019). The cave site receives the majority of its high annual precipitation (∼2500 mm yr$^{-1}$) from eastward moving air masses originating over the North Atlantic Ocean, resulting in a coupling between large-scale surface ocean conditions in the North Atlantic and geochemical signals preserved in speleothems at the site (e.g. Drysdale et al., 2020). Owing to this linkage, Corchia Cave has yielded an abundance of palaeoclimate data of regional and global significance, including precise constraints on the timing of glacial terminations (Drysdale et al., 2020; Bajo et al., 2020) and interglacial climate variability (Regattieri et al., 2014; Tzedakis et al., 2018).

Palaeoclimate reconstruction studies undertaken at Corchia Cave have largely focussed on stalagmites collected from the large and well-decorated Galleria delle Stalattiti chamber. This chamber sits approximately 400 m below the surface, at an elevation

---

[1]Where $\delta^{238}\mathrm{U} = \dfrac{\left(\frac{^{238}\mathrm{U}}{^{235}\mathrm{U}}\right)_{\mathrm{sample}}}{\left(\frac{^{238}\mathrm{U}}{^{235}\mathrm{U}}\right)_{\mathrm{CRM-112A}}} - 1$. Note that the CRM-112A reference material has an equivalent isotopic composition to CRM-145.

of 835 m above sea level, and about 800 m from the nearest natural cave opening (Piccini et al., 2008). The microclimate of Galleria delle Stalattiti is relatively stable, typical of deep-cave environments, with a humidity consistently above 98% and an average present-day temperature of ~7.9 °C (Drysdale et al., 2019). Stalagmites from this chamber are typically well suited to U-series geochronology with consistently high U content (typically 3–20 ppm) and very low detrital Th, as well as relatively low inherited Pb (Bajo et al., 2012). Furthermore, most stalagmites are composed of clean (i.e. free of visible detritus) compact columnar calcite, which is typically associated with the preservation of primary geochemical signals and closed-system behaviour with respect to U-series nuclides (Scholz et al., 2014). These characteristics make stalagmites from the Galleria delle Stalattiti very well-suited to U-series geochronology, and when combined with the site's suitability for palaeoclimate reconstruction, have led to a large number of U–Th and U–Pb determinations being undertaken on speleothems from the site. These data suggest that initial $^{234}$U/$^{238}$U values of speleothems from this cave chamber are always less than 1 and have undergone a long-term evolution over the past several hundred kyr towards more $^{234}$U-depleted values (Woodhead et al., 2006), with activity ratios as low as ~0.65 obtained for Late Holocene materials (Zanchetta et al., 2007).

This study focusses on three stalagmites (sample IDs: CCB, CC2/CC15, and CC17-1) that were found broken near their original in situ position during field campaigns between 2001 and 2017. These stalagmites are large, with lengths of 67–130 cm, and average diameters ranging from approximately 9–22 cm. Following collection, the stalagmites were halved along their vertical growth axes and polished to aid in identification of visible growth laminations. In section, the stalagmites are white to grey in colour and predominately composed of compact translucent-to-opaque primary calcite. Exceptions are the very top of CCB, which contains some slightly porous calcite as the diameter tapers toward the tip, and the middle of CC17-1, which contains some localised, slightly porous sections associated with sharp changes in growth axis direction. These features are atypical of Galleria delle Stalattiti speleothems and were avoided during sampling.

## 3 Analytical methods

### 3.1 Sampling

Samples for both U–Th and U–Pb dating were extracted using a dental drill, targeting clean and compact calcite by carefully tracing along visible growth contours approximately 5 mm apart at the central axis but tapering along the flanks. Several solid sub-samples of ~4 mm (width) × ~4 mm (length) × ~3 mm (depth), or 20–120 mg, were then extracted from each growth domain. For each sampling position, one sub-sample, taken from the centre of the sampling area, was used for U–Th analysis and the remaining 4–11 sub-samples were used for U–Pb analysis. The only exception to this is sample CC17-1-3, where the U–Th sample was taken from a growth layer marginally below that of the corresponding U–Pb samples. However, this sampling offset was negligible relative to the age of these growth layers.

## 3.2 U–Th geochronology

U and Th isotopic analysis was performed at the University of Minnesota using a modified version of the all Faraday cup mass spectrometry protocol of (Cheng et al., 2013), but with the $^{234}U^+$ and $^{230}Th^+$ ion beams collected in a Faraday cup fitted with a $10^{13}$ $\Omega$ resistor (Pythoud, 2022). Chemistry procedures were similar to those described in (Edwards et al., 1987). Briefly, solid samples of 20–110 mg (equivalent to $\sim$500 ng U) were dissolved in $\sim$1% $HNO_3$ before addition of a mixed $^{233}U$-$^{236}U$-$^{229}Th$ tracer calibrated against IRMM-074/10 (Cheng et al., 2013). Approximately six drops of concentrated $HClO_4$ were added to each sample beaker before being capped and refluxed on a hotplate for several hours to ensure complete sample-spike equilibration. Following complete dry down, separation and purification of U and Th proceeded in two stages. Firstly, U and Th were separated from the matrix using Fe co-precipitation in 2 N HCl by drop-wise addition of $NH_4OH$ followed by centrifuging. Secondly, U and Th were separated and further purified via elution chromatography using BioRad AG1-X8 anion exchange resin, with the Th fraction collected in 6 N HCl and the U fraction collected in ultra-pure water. The Th fraction was then passed through a second identical column to ensure complete separation from U. Following this, the purified U and Th sample fractions were dried down and sequentially treated with $HClO_4$ and $HNO_3$ to oxidise any remaining organics, before being taken up in 1% $HNO_3$ for analysis.

The U and Th isotope ratios were measured separately in static mode on a Thermo-Scientific Neptune Plus MC-ICPMS (see The Supplement for details on the detector configuration). Samples were introduced into the mass spectrometer via a Cetac Aridus II desolvating-nebulising system fitted with a C50 PFA concentric nebuliser, using an uptake rate of 30–50 $\mu$L min$^{-1}$. Zeros were measured for 13 minutes for both U and Th immediately prior to sample introduction. U measurements were then made over 5–10 minutes with a $^{234}U^+$ signal size between 8–18 mV. Th measurements immediately followed U for each sample and were made over 1–5 minutes with a $^{230}Th^+$ signal size between 8–20 mV.

For Faraday cups fitted with $10^{10}$ $\Omega$ and $10^{11}$ $\Omega$ resistors, amplifier gain calibration was performed via the mass spectrometer software using a standard current at the start of each run. However, the limited dynamic range of the $10^{13}$ $\Omega$ resistor precluded use of this approach for the cup fitted with this resistor. Therefore, amplifier gain on this detector was characterised by periodic ($\sim$fortnightly) measurement of Nd standard SRM 3135a following Pythoud (2022). Tailing of the $^{238}U^+$ and $^{235}U^+$ ion beams was characterised by analysis of un-spiked CRM-112A prior to each run, with the background measured at masses 233, 233.5, 234.5, 236 and 237 amu on the axial SEM with the Retarding Potential Quadrupole (RPQ) off. A logarithmic regression fit through these data was then used to estimate tailing onto the mass 233–236 cups as a function of the signal intensity at mass 237 during measurement (Pythoud, 2022). For each U fraction, tailing was corrected by measuring the background at mass 237 on the axial SEM during sample analysis and applying the predicted $M_{tail}/237$ ratios (where $M$ refers to masses 233–236). Tailing of the $^{232}Th^+$ beam onto the mass 230 cup was negligible given the very high $^{230}Th/^{232}Th$ ratios of the samples analysed in this study and was not corrected for.

**Table 1.** U–Th dating results.

| Sample ID | U (ppm) | $\delta^{234}U^{1}$ (‰) | $\left[^{230}\text{Th}/^{238}\text{U}\right]$ | $\left[^{230}\text{Th}/^{232}\text{Th}\right]$ | Age (ka)[2] | Age 95% CI (ka)[3] | $\delta^{234}U_{\text{initial}}$ (‰) |
|---|---|---|---|---|---|---|---|
| CCB-B-1 | 18.05 | $-85.94\pm0.32$ | $0.86499\pm0.00034$ | $1.30\times10^{6}$ | $433.3\pm4.5$ | (429.0, 437.8) | $-291.9\pm4.6$ |
| CCB-C-3 | 8.10 | $-84.01\pm0.28$ | $0.86925\pm0.00037$ | $1.96\times10^{5}$ | $447.8\pm5.0$ | (443.0, 452.8) | $-297.3\pm4.9$ |
| CCB-C-1 | 7.23 | $-80.93\pm0.29$ | $0.87390\pm0.00041$ | $1.17\times10^{5}$ | $452.1\pm5.5$ | (446.9, 457.7) | $-289.9\pm5.2$ |
| CCB-C-20 | 11.36 | $-80.10\pm0.25$ | $0.87532\pm0.00026$ | $2.22\times10^{6}$ | $455.0\pm4.3$ | (450.9, 459.3) | $-289.3\pm4.3$ |
| CC17-1-1 | 5.11 | $-91.45\pm0.28$ | $0.86025\pm0.00046$ | $1.78\times10^{5}$ | $459.4\pm6.6$ | (453.2, 466.1) | $-334.4\pm6.9$ |
| CC17-1-35 | 3.51 | $-82.39\pm0.29$ | $0.87545\pm0.00027$ | $7.73\times10^{4}$ | $495.0\pm7.4$ | (488.0, 502.5) | $-333.0\pm7.9$ |
| CCB-E-9 | 7.53 | $-66.94\pm0.32$ | $0.89800\pm0.00028$ | $3.19\times10^{5}$ | $513.4\pm9.0$ | (505.0, 522.6) | $-285.1\pm8.5$ |
| CCB-E-10 | 6.84 | $-63.60\pm0.26$ | $0.90340\pm0.00022$ | $9.56\times10^{5}$ | $527.0\pm8.2$ | (519.2, 535.4) | $-281.4\pm7.6$ |
| CCB-F-16 | 6.96 | $-61.39\pm0.27$ | $0.90664\pm0.00030$ | $4.00\times10^{5}$ | $530.1\pm9.4$ | (521.3, 539.8) | $-274.0\pm8.3$ |
| CCB-F-1 | 5.38 | $-60.26\pm0.31$ | $0.90843\pm0.00039$ | $5.67\times10^{5}$ | $534.4\pm11.6$ | (523.7, 546.5) | $-272.3\pm10.0$ |
| CC2-1 | 7.10 | $-62.32\pm0.18$ | $0.90650\pm0.00026$ | $2.04\times10^{6}$ | $555.6\pm9.2$ | (547.0, 565.0) | $-298.9\pm8.4$ |
| CCB-6-1 | 7.43 | $-50.75\pm0.37$ | $0.92490\pm0.00026$ | $1.01\times10^{6}$ | $628.0\pm29.7$ | (603.3, 661.5) | $-298.6\pm27.4$ |
| CC17-1-3 | 4.53 | $-57.57\pm0.35$ | $0.91440\pm0.00033$ | $2.97\times10^{5}$ | $592.2\pm21.6$ | (573.4, 615.8) | $-306.2\pm20.4$ |
| CC15-1 | 3.36 | $-48.70\pm0.34$ | $0.92789\pm0.00037$ | $7.14\times10^{5}$ | $632.4\pm32.1$ | (606.0, 669.0) | $-290.1\pm28.5$ |

Uncertainties are given at the $2\sigma$ level unless otherwise specified. Decay constant values: $\lambda_{238} = 1.55125 \times 10^{-10}$ (Jaffey et al., 1971) and $\lambda_{234} = 2.82206 \times 10^{-6}$ (Cheng et al., 2013), and $\lambda_{230} = 9.1705 \times 10^{-6}$ (Cheng et al., 2013). [1]$\delta^{234}U = \left[^{234}U/^{238}U\right] - 1$. [2]Uncertainties calculated by first-order algebraic uncertainty propagation (Appendix B). [3]Calculated by Monte Carlo simulation.

Data reduction was performed using an offline spreadsheet that implements a tau correction following Pythoud (2022) to account for the slow response time of $10^{13}$ $\Omega$ resistors, before applying the $^{238}$U tail correction as described above, and mass bias correction based on the exponential law (Russell et al., 1978) using the $^{233}$U/$^{236}$U ratio measured for each sample. The mass bias factor calculated for the U fraction was subsequently used to correct the Th fraction run immediately after, based on the assumption of near-equivalent mass bias behaviour between U and Th isotopes (Potter et al., 2005). This is justified by the consistent mass bias behaviour observed on this instrument, especially over multiple successive analyses (Pythoud, 2022). Finally, corrected U ratios were normalised to CRM-112A, which was periodically analysed throughout each run, using the value of $52.852 \pm 0.015$ obtained by Cheng et al. (2013). U–Th ages and uncertainties (Table 1) were then computed using a Python script that calculates ages and age uncertainties using both first-order algebraic uncertainty propagation (Appendix A1) and Monte Carlo approaches (Hellstrom, 1998; Ludwig, 2003) (see The Supplement for further details).

Owing to the very high $^{230}$Th/$^{232}$Th ratios of the samples analysed in this study, we opted not to correct U–Th or U–Pb ages for initial $^{230}$Th in order to simplify age and age uncertainty calculations (i.e. Appendix B). We note, however, that if such a correction were applied it would result in a negligible correction of at most a few years for both U–Th and U–Pb ages.

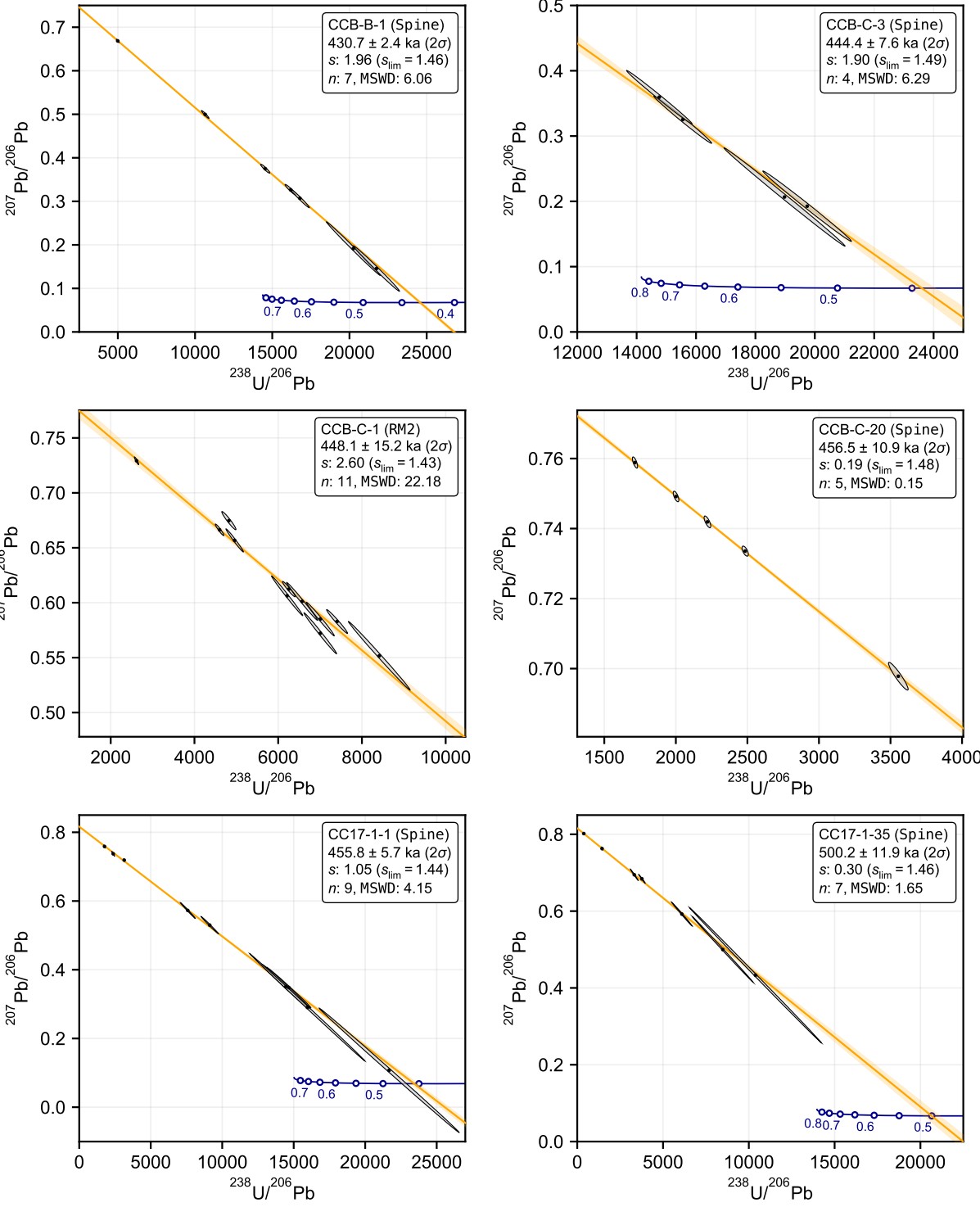

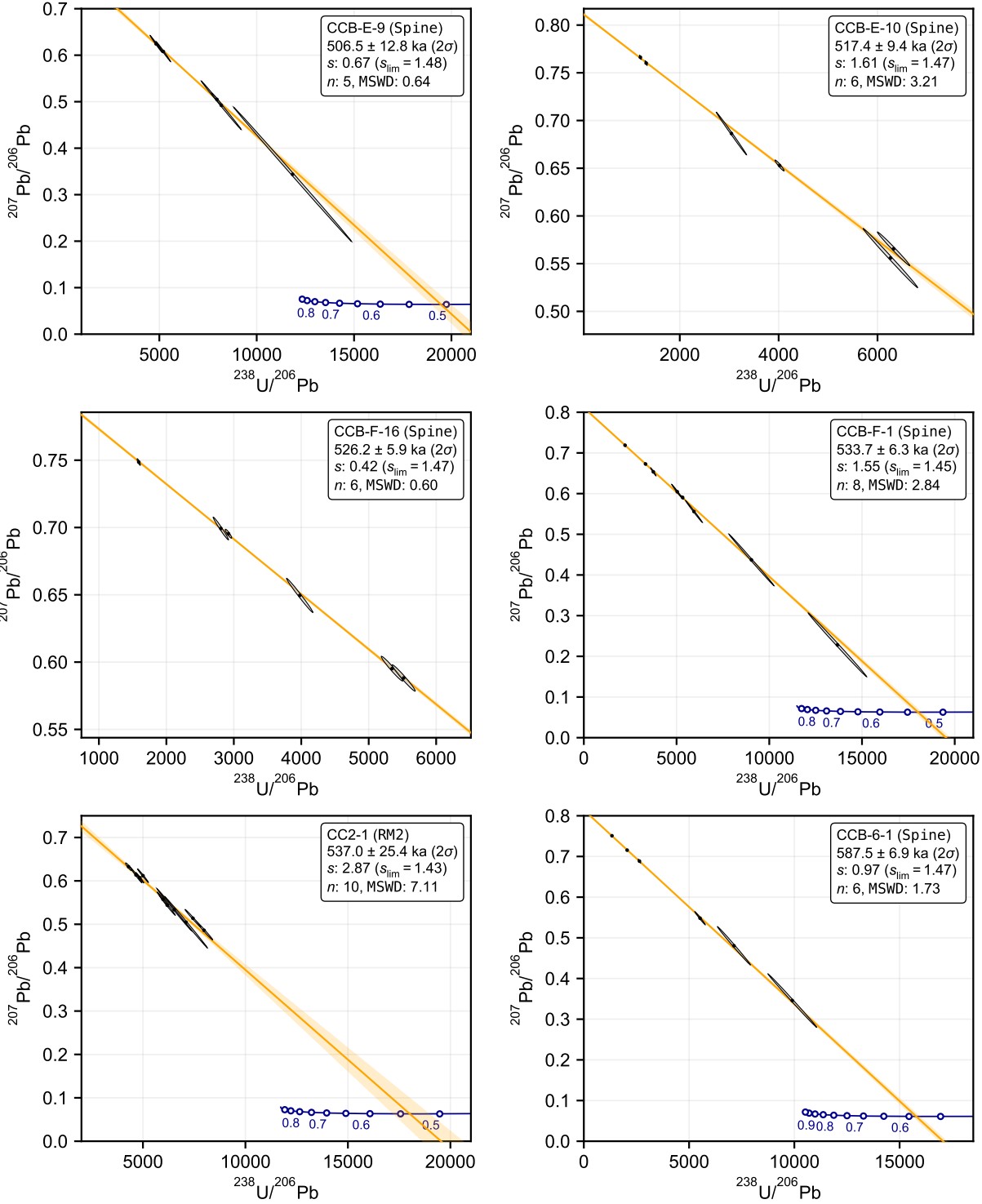

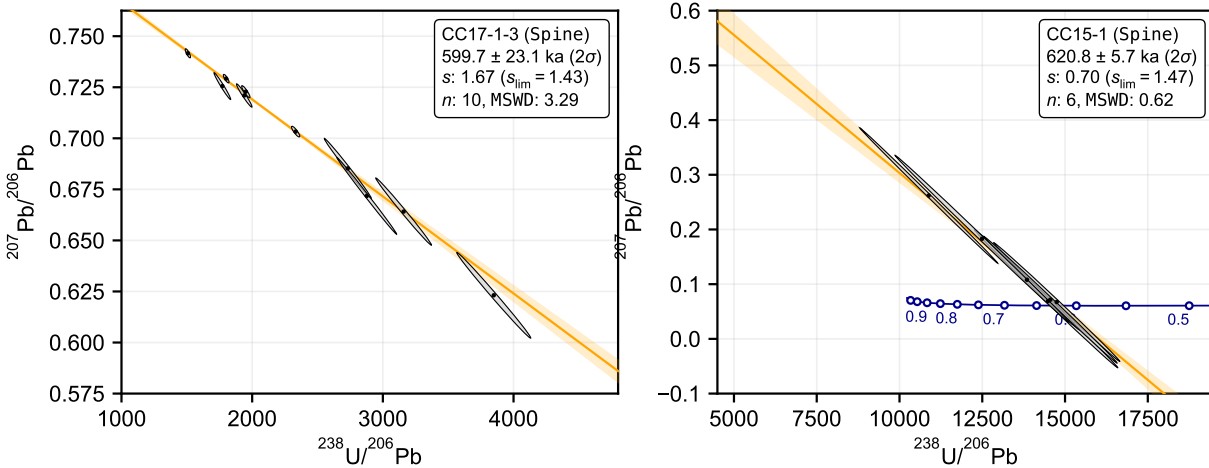

**Figure 1.** U–Pb Tera-Wasserburg isochron diagrams for the samples analysed in this study. Measured data points are plotted as 95% confidence ellipses (grey) along with the fitted isochron lines (orange) and their 95% confidence bands (orange shading). The dark blue line shows the disequilibrium concordia curve, constructed using the measured $\left[^{234}U/^{238}U\right]$ value and assuming $\left[^{230}Th/^{238}U\right] = \left[^{226}Ra/^{238}U\right] = \left[^{231}Pa/^{235}U\right] = 0$. This Concordia curve is truncated at the youngest age point associated with a physically impossible negative initial activity ratio solution following McLean et al. (2016). Shown in the textboxes is the `spine` width, $s$, value for a `spine` regression fit and simulated upper 95% confidence limit on $s$ (see text for further discussion). The `MSWD` (mean square weighted deviation) for a classical weighted least-squares regression (e.g. York et al., 2004) is also provided for comparison with other data sets. The regression model implemented is indicated in brackets: `spine` indicates use of the algorithm described by Powell et al. (2020), `RM2` indicates use of the Robust Model 2 algorithm (Pollard et al., 2023).

### 3.3 U–Pb geochronology

U–Pb dating was performed at the University of Melbourne. Analytical methods built on those published previously (Woodhead et al., 2012; Sniderman et al., 2016) but using the streamlined chemistry procedure described in Engel et al. (2020). Solid calcite samples were repeatedly cleaned by brief ($\sim$1 min) immersion in dilute ($\sim$0.01 M) multiply-distilled HCl, followed by washing in ultra-pure water, before drying in a HEPA-filtered laminar flow hood. The cleaned samples were then individually weighed into pre-cleaned Teflon beakers and treated with sufficient 6 N HCl to ensure complete dissolution. A mixed $^{233}U/^{205}Pb$ isotopic tracer, calibrated against EarthTime reference solutions (Condon et al., 2015), was then weighed into the vials and each one sealed and refluxed on the hotplate for several hours to ensure complete sample-spike equilibration. Samples were then dried down and U and Pb separated from the calcite matrix using the single-column mixed-resin ion-exchange technique described in Engel et al. (2020).

Isotope ratios were measured on a Nu Plasma MC–ICP-MS using a DSN-100 desolvation unit and MicroMist glass nebuliser, operating with an uptake rate of $\sim$80 $\mu$L min$^{-1}$. Instrumental mass bias effects were monitored and corrected using NIST SRM 981 reference material in the case of Pb, and the sample's internal $^{238}$U/$^{235}$U ratio in the case of U.

U–Pb isochron regression lines were fitted using DQPB (Pollard et al., 2023), initially employing the spine robust isochron algorithm (Powell et al., 2020). This algorithm accounts for analytical uncertainties and can accommodate datasets with weighted residuals that depart from a strict Gaussian distribution but converges to the classical least-squares solution of York et al. (2004) for 'well behaved' datasets (i.e. those where data scatter is accounted for by assigned measurement uncertainties). The spine-width, $s$, was used to assess suitability of this algorithm for each dataset in relation to the upper 95% confidence limit on $s$ (here denoted $s_{\text{lim}}$) obtained by simulation of Gaussian distributed datasets (Powell et al., 2020).

In cases where $s$ clearly exceeded $s_{\text{lim}}$, the Robust Model 2 (RM2) algorithm presented in Pollard et al. (2023) was implemented instead. This algorithm ignores analytical uncertainties, inferring a robust dispersion scale from observed data scatter itself rather than assigned measurement uncertainties. Therefore, it is arguably better suited to data sets containing significant overdispersion (i.e. data scatter in excess of that attributable to assigned measurement uncertainties). However, for isochrons where $s$ only marginally exceeded $s_{\text{lim}}$, the spine algorithm was considered likely to provide a more accurate best-fit line than the RM2 algorithm, although in this case the spine uncertainty calculations become potentially unreliable. Thus, where $s$ exceeded $s_{\text{lim}}$ by less than an arbitrary but reasonable value of 50% the spine regression line was retained, but uncertainties were expanded in an attempt to accommodate data scatter above what is reasonably accounted for by measurement uncertainties. This was achieved by multiplying the covariance matrix of the slope and y-intercept, $\mathbf{V}_\theta$, by $\left(s/s_{\text{lim}}\right)^2$. Implementing the RM2 algorithm instead for these datasets does not significantly alter any of the conclusions of this study.

U–Pb ages were calculated as the intercept point between the regression line and disequilibrium concordia curve on a Tera-Wasserburg diagram (Fig. 1), incorporating the measured $\left[^{234}\text{U}/^{238}\text{U}\right]$ value from the U–Th analysis into the iterative age-solving procedure (Appendix A2), and assuming initial $\left[^{230}\text{Th}/^{238}\text{U}\right]$, $\left[^{231}\text{Pa}/^{235}\text{U}\right]$, $\left[^{226}\text{Ra}/^{238}\text{U}\right]$ values equal to 0. Uncertainties were calculated using a Python script that implements the Monte Carlo simulation approach described in (Pollard et al., 2023) with $10^6$ iterations (see The Supplement for further details) and also via first-order algebraic uncertainty propagation (Appendix B).

For the purpose of determining analytical accuracy, each U–Pb session included analysis of several 'synthetic zircon' standards: ET-100Ma, ET-500Ma, and ET-2Ga. These materials, produced by the EarthTime initiative, have nominal U and Pb compositions approximating zircons with the ages of 100 Ma, 500 Ma, and 2 Ga, but without any matrix (Condon et al., 2008). While they were originally conceived for quality control during TIMS single zircon analysis, and are still routinely used for that purpose (Szymanowski and Schoene, 2020), they are also ideally suited to isotope dilution (ID) carbonate U–Pb studies since the quantities of U and Pb being analysed can be controlled to approximate unknowns and the highly radiogenic nature of the Pb involved serves to rapidly identify any anomalous analytical issues, such as unexpected high blanks. In our studies, we pass

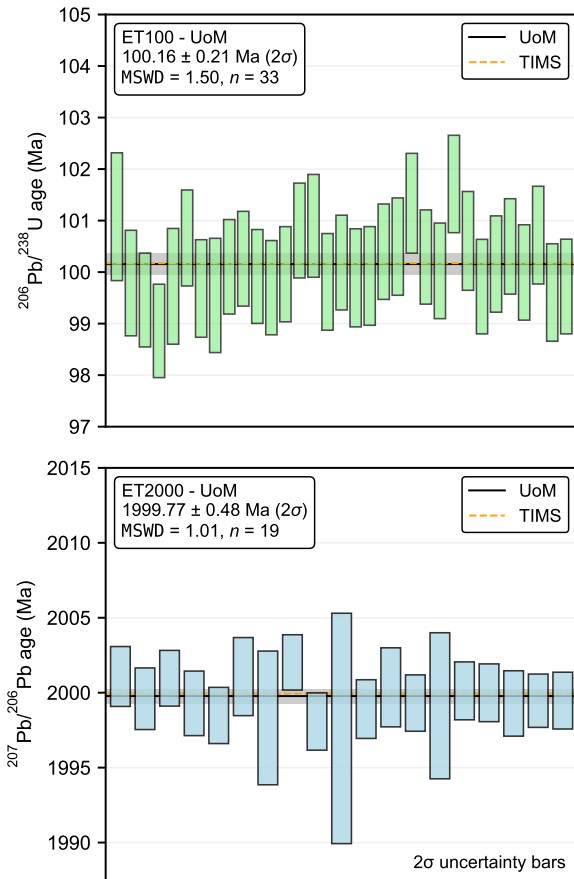

**Figure 2.** Radiogenic $^{206}Pb/^{238}U$ and $^{207}Pb/^{206}Pb$ age determinations for EarthTime (ET) 'synthetic zircon' standards analysed during the course of this study at the University of Melbourne (UoM) using a similar Pb load to the speleothem samples and identical column chemistry and mass spectrometry procedures. The weighted average age is plotted as the black line with grey shading indicating the 95% confidence band. Also shown for comparison are the average inter-laboratory TIMS ages obtained by Schaltegger et al. (2021). $^{207}Pb/^{206}Pb$ ages were calculated using $^{238}U/^{235}U = 137.818$.

255 aliquots of these solutions (typically equating to ∼5 ng of radiogenic Pb) through the same column chemistry as our samples. The measured isotopic ratios of these standards analysed during the course of this study were consistent with benchmark TIMS values (Fig. 2).

## 4 Results and discussion

### 4.1 Age consistency

260 We first evaluated U–Th and U–Pb age consistency through a pairwise comparison. For each sample, we tested whether the U–Th and U–Pb ages agreed within their analytical uncertainties using a hypothesis test that amounted to assessing if their age difference was statistically consistent with zero (e.g. Barlow, 1989). The test statistic was calculated as

$$z = \frac{|\Delta|}{\sigma_\Delta} \tag{1}$$

such that

265 $$\Delta = t_{Th} - t_{Pb} \tag{2}$$

where $t_{Th}$ and $t_{Pb}$ represent the U–Th and U–Pb ages and $\sigma_\Delta$ is the standard error of the age difference, calculated via first-order uncertainty propagation as

$$\sigma_\Delta = \sqrt{\sigma_{t_{Th}}^2 + \sigma_{t_{Pb}}^2 - 2\sigma_{t_{Th}}\sigma_{t_{Pb}}\rho} \tag{3}$$

Here, $\sigma_{t_{Th}}$ and $\sigma_{t_{Pb}}$ are the standard errors of the U–Th and U–Pb ages and $\rho$ is their correlation coefficient arising from shared 270 $\left[^{234}U/^{238}U\right]$ uncertainties (see Appendix B). However, in most cases this correlation term was relatively minor, and negligible in cases where U–Pb regression fitting uncertainties were relatively large. We then obtained a formal $p$-value under the null hypothesis that there is no significant difference between the ages by comparing $z$ against the standard normal distribution.

Strictly speaking, this approach is only appropriate if uncertainties in both U–Th and U–Pb ages conform to a Gaussian distribution. While this assumption is reasonable for the majority of ages, it is slightly inaccurate for the older U–Th ages (i.e. 275 samples CCB-6-1, CC17-1-3, and CC15-1), where the age uncertainty distributions are positively skewed owing to the non-linearity of the U–Th age equation, which becomes more apparent for older ages and those with larger analytical uncertainties (e.g. Ludwig, 2003). Therefore, for these samples, we instead implemented a Monte Carlo procedure to simulate the age difference distribution (see The Supplement for details). Analogous to the formal hypothesis test, we concluded that the age difference was consistent with zero if the estimated 95% confidence interval on the Monte Carlo simulated age difference 280 overlapped zero. The results of this pairwise age comparison are given in Table 2.

Our results show excellent agreement between U–Th and U–Pb ages within analytical uncertainties for all samples (Fig. 3), with the exception of sample CCB-6-1 where the U–Th age is significantly older than the U–Pb age by 41 +34/-25 ka (95% confidence). In this particular case, we consider the U–Pb age to be more reliable for two key reasons. Firstly, the U–Pb isochron is well formed, with good spread in U/Pb ratios and data scatter that is consistent with analytical uncertainties 285 ($s < s_{\mathrm{lim}}$ and $p =$0.14 for a classical least-squares fit). Secondly, the U-Pb age better aligns with the sample's stratigraphic position when ages from all three stalagmites are aligned to a common depth scale by synchronising their carbon ($\delta^{13}$C)

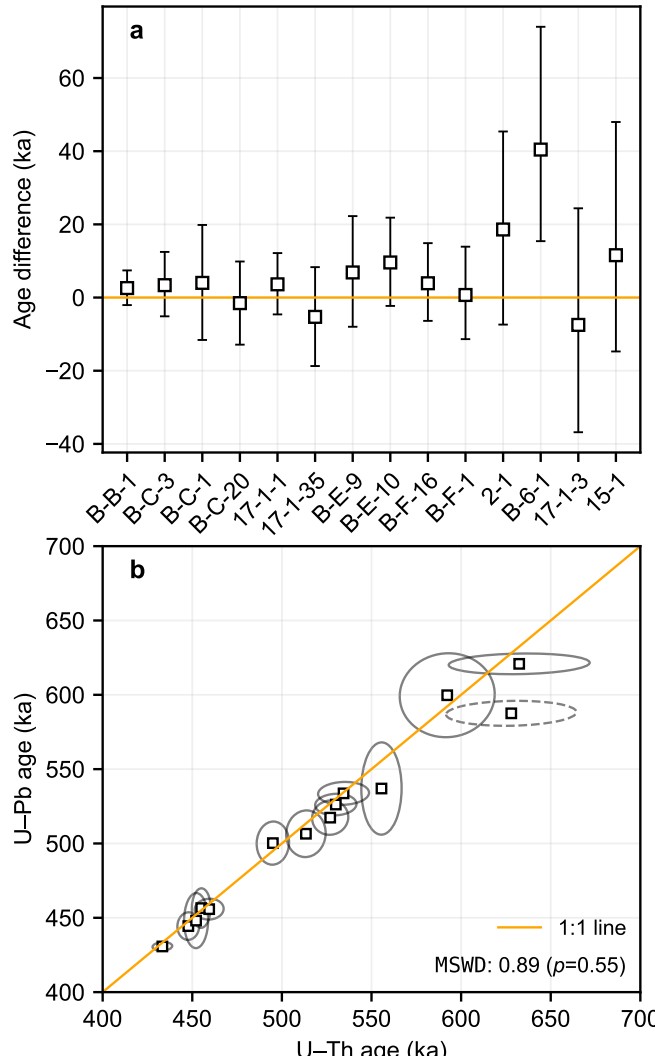

**Figure 3.** Comparison of U–Th and U–Pb age determinations. (a) Age difference for each sample calculated as: $t_{Th} - t_{Pb}$. Note that the 'CC' prefix of each sample ID has been omitted for brevity. (b) Collective comparison U–Th and U–Pb ages. The confidence ellipse for sample CCB-6-1, which was found to be inconsistent on a pairwise basis and therefore excluded from the MSWD calculation, is plotted with the dashed outline. Error bars and confidence ellipses are plotted at the 95% confidence level.

and oxygen ($\delta^{18}$O) stable isotope profiles—noting that there is generally very good agreement between isotopic variations amongst coeval speleothems from the Galleria delle Stalattiti chamber of Corchia Cave (Tzedakis et al., 2018; Bajo et al., 2020). The cause of the anomalously old U–Th age is uncertain, but one possibility is that this particular sub-sample was affected by localised open-system behaviour which resulted in post-depositional U loss and thus an older-than-true U–Th age determination. This is known to affect some speleothem samples even if they visually appear pristine (Bajo et al., 2016).

**Table 2.** Summary of pairwise U–Th versus U–Pb age comparison

| Sample ID | Age (ka)[1] | Age diff. (ka)[2] | Age diff. 95% CI (ka)[4] | $p$-value[5] | Significant difference?[6] |
|---|---|---|---|---|---|
| CCB-B-1 | 431.2 | 2.6 | (-2.1, 7.4) | 0.28 | no |
| CCB-C-3 | 446.8 | 3.4 | (-5.1, 12.5) | 0.45 | no |
| CCB-C-1 | 451.7 | 4.0 | (-11.6, 19.8) | 0.62 | no |
| CCB-C-20 | 455.2 | -1.5 | (-12.9, 9.9) | 0.79 | no |
| CC17-1-1 | 457.4 | 3.6 | (-4.6, 12.2) | 0.40 | no |
| CC17-1-35 | 496.4 | -5.3 | (-18.7, 8.3) | 0.44 | no |
| CCB-E-9 | 511.2 | 6.9 | (-8.0, 22.2) | 0.37 | no |
| CCB-E-10 | 522.9 | 9.6 | (-2.3, 21.9) | 0.12 | no |
| CCB-F-16 | 527.2 | 3.9 | (-6.4, 14.9) | 0.47 | no |
| CCB-F-1 | 533.9 | 0.7 | (-11.4, 13.9) | 0.91 | no |
| CC2-1 | 553.5 | 18.6 | (-7.4, 45.4) | 0.17 | no |
| CCB-6-1 | 588.9 | 40.4 | (15.4, 74.0) | 0.01* | yes |
| CC17-1-3 | 595.7 | -7.5 | (-36.8, 24.5) | 0.63 * | no |
| CC15-1 | 621.0 | 11.6 | (-14.8, 48.0) | 0.45* | no |

[1] Weighted average of U–Th and U–Pb ages. [2] Age difference calculated as: $t_{Th} - t_{Pb}$. [3] Calculated by first-order algebraic uncertainty propagation. [4] Calculated by Monte Carlo simulation. [5] p-value under the null hypothesis that there is no significant age difference. [6] Result of the hypothesis test of age consistency at the 95% confidence level. *p-value should be interpreted as a qualitative indicator only (see text).

To evaluate collective agreement between U–Th and U–Pb ages, we also employed linear regression approach. For two consistent and unbiased dating methods, paired age determinations should plot along a 1:1 line passing through the origin within their analytical uncertainties. After excluding sample CCB-6-1, which was already found to be inconsistent on a pairwise basis,

we fitted a regression line through the remaining paired ages using the weighted least-squares algorithm of York et al. (2004), which accounts for assigned uncertainties in both variables and uncertainty correlation (Fig. 3b). The best-fit regression line had a slope of $0.98 \pm 0.07$ ($2\sigma$), a y-intercept value of $6.0 \pm 31$ ($2\sigma$), and an `MSWD` of 0.50 (n=14, p=0.74). We also assessed goodness of fit to a 1:1 line passing through the origin directly, obtaining a `MSWD` of 0.89 (n=14, p=0.55). These results suggest that the two dating methods are consistent and unbiased over the age interval considered in this study.

One limitation of this approach, however, is that—as noted above—the U–Th age analytical uncertainties for older samples CC15- 1 and CC17-1-3 are slightly non-Gaussian, leading to some inaccuracy in regression results obtained by the classical least-squares approach, which assumes strictly Gaussian uncertainty distributions. Nevertheless, these samples have relatively large age uncertainties that comfortably overlap the best-fit line; we thus consider the results to be relatively insensitive to this assumption.

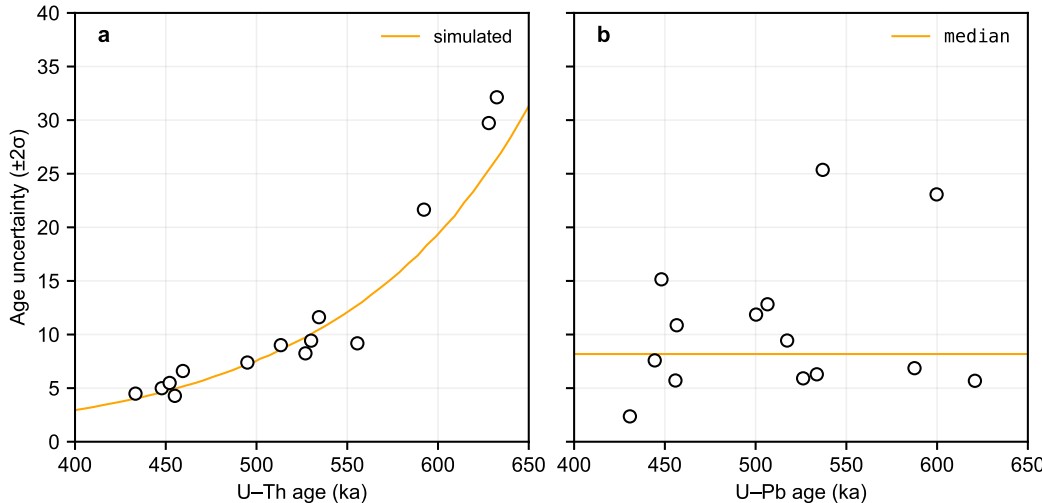

**Figure 4.** Age precision versus age for U–Th and U–Pb age determinations The orange line in (a) shows the predicted U–Th age uncertainty calculated using Monte Carlo simulation and based on the average $[^{234}U/^{238}U]$ and $[^{230}Th/^{238}U]$ measurement uncertainties of samples analysed in this study and an assumed $[^{234}U/^{238}U]_i$ value of 0.72, which is typical of Corchia Cave stalagmites falling in this age range. The orange line in (b) shows the median U–Pb age uncertainty over this interval.

## 4.2 Age uncertainties

The uncertainty characteristics of U–Th and U–Pb ages differ significantly over the age interval considered in this study. Overall, U–Th age uncertainties show more predictable behaviour than the U–Pb age uncertainties and tend to increase in an approximately exponential manner from ∼4.5 ka ($2\sigma$) at ∼450 ka to >25 ka ($2\sigma$) at ∼620 ka (see Fig. 4). When U–Th age uncertainties are plotted against U–Th age, the data tend to follow a prediction trend based on propagating average

$[^{234}U/^{238}U]$ and $[^{230}Th/^{238}U]$ analytical uncertainties obtained in this study through to U–Th age uncertainties. Departures from this trend primarily reflect deviations in analytical uncertainties of one or both of the measured ratios from the average. While the exact trajectory of this line is expected to vary for different analytical protocols, the exponential form is not, provided that initial $^{230}Th$ is either negligible or reasonably consistent. The trajectory of the line may also depend somewhat on initial $[^{234}U/^{238}U]$ values of the samples, with lower precision ages being obtained for samples with $[^{234}U/^{238}U]$ values <1 (e.g

Meckler et al., 2012), all else being equal.

U–Pb age uncertainties, on the other hand, do not show an obvious trend with increasing age and do not correlate directly with average analytical uncertainties in the $^{238}U/^{206}Pb$ or $^{206}Pb/^{207}Pb$ isotope ratio measurements. Moreover, there is only a relatively small correlation between U–Pb age uncertainties in this dataset and $[^{234}U/^{238}U]$ measurement precision or scatter of data about the isochron, quantified either by the MSWD statistic or its robust variant, $s$ (see The Supplement). Instead, U–Pb

age uncertainties appear to depend predominantly on the distribution of data along the isochron, consistent with the findings of previous studies (Woodhead et al., 2012; Engel and Pickering, 2022).

Engel and Pickering (2022) examined factors controlling U–Pb isochron age precision in Middle Pleistocene speleothems from multiple cave sites, including Corchia Cave. They found that uncertainty in Tera-Wasserburg U–Pb isochron ages generally correlates strongly with uncertainty in the isochron regression slope, which is in-turn controlled largely by spread in Pb/U ratios

along the isochron line due to variability in inherited-Pb content. To quantify the spread of data points along the isochron, Engel and Pickering (2022) defined a metric termed 'average isochron distance', which quantifies the spread of data as the average Euclidean distance between each data point and the centroid of the data set $(\bar{x}, \bar{y})$. Here we adopt an alternative metric based on the least-squares fitted points[2] along the regression line rather than the measured data points themselves. In this context, use of the least-squares fitted points helps to better differentiate between spread of data points *along* the isochron and scatter

*about* the isochron (e.g. due to 'geological' scatter). It also circumvents scaling issues associated with isochron diagram axes spanning vastly different orders of magnitude, as is the case for the Tera-Wasserburg diagram. The metric we adopt is termed 'combined isochron spread' and is calculated as

$$d_x = \sqrt{\sum_k (x_k - \bar{x})^2} \tag{4}$$

where $x_k$ is the least-squares fitted $^{238}U/^{206}Pb$ value of the $k^{\text{th}}$ data point and $\bar{x}$ is the centroid of the data points.

In addition to spread of data points along the isochron, another factor that influences Tera-Wasserburg U–Pb isochron age precision is the location of the data relative to the concordia intercept point. To assess this effect on U–Pb isochron age uncertainties, we define a second metric termed the 'average inherited Pb index', which is calculated as:

$$\bar{Pb}_i = \frac{\bar{y} - y^*}{y_0 - y^*} \tag{5}$$

where $y^*$ is the $^{207}Pb/^{206}Pb$ value of the concordia intercept point, $\bar{y}$ is the centroid of the data, and $y_0$ is the y-axis intercept

point (i.e. the estimated inherited $^{207}Pb/^{206}Pb$ value from the regression fit). Essentially, this is the distance in $y$ between the concordia intercept point and the centroid of the data, normalised to the maximum possible distance that could be obtained for a given age, initial $^{207}Pb/^{206}Pb$ isotopic composition, and initial $\left[^{234}U/^{238}U\right]$ value.

Consistent with Engel and Pickering (2022), we find that the spread of data along the isochron line is controlled mostly by variation in inherited-Pb content rather than U content, with the latter tending to remain relatively constant across speleothem

growth layers (see Fig. S4 in The Supplement). We find that U–Pb isochron age uncertainties correlate strongly with uncertainty in the isochron slope, excluding the highly radiogenic (i.e. very low inherited Pb) samples CC15-1 and CCB-C-3, which

---

[2]The least-squares fitted points (referred to as 'least-squared adjusted' points by York et al., 2004) are the points along the regression line with the highest probability of generating the 'uncertainty perturbed' measured data points assuming that assigned analytical uncertainties are the only reason measured values depart from the isochron line.

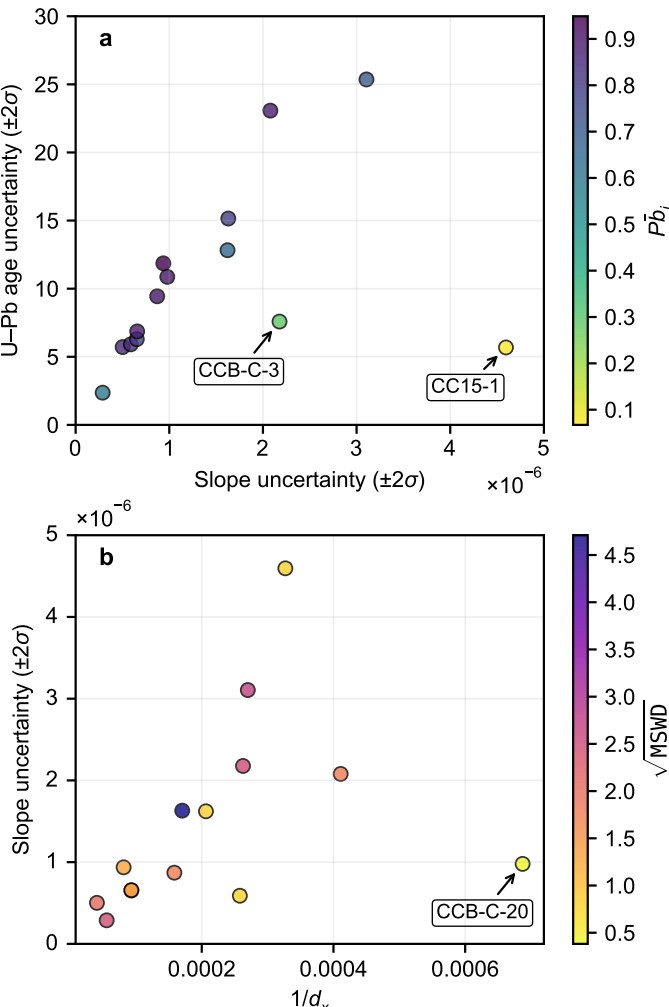

**Figure 5.** Factors affecting U–Pb isochron age precision. (a) U–Pb age uncertainty plotted against uncertainty in the isochron slope. Markers are coloured according to $\bar{P}b_i$ (i.e. the 'inherited Pb index', Eq. 5). Age uncertainty generally correlates strongly with uncertainty on the isochron slope, except where isochrons are highly radiogenic (i.e. have very low average inherited Pb). (b) Isochron slope uncertainty plotted against the reciprocal of the 'combined isochron spread' ($d_x$) metric (Eq. 4). Generally speaking, isochron slope uncertainty is correlated with the inverse of combined Pb/U spread. An obvious exception is sample CCB-C-20, which exhibits very little data scatter about the isochron. Markers are coloured according to $\sqrt{\texttt{MSWD}}$, employed here as an indicator of data dispersion about the isochron.

encompass multiple data points plotting close to, or overlapping, the disequilibrium concordia curve (Fig. 5a). In turn, isochron slope precision correlates with combined isochron spread (Fig. 5b), although sample CCB-C-20, which has an exceptionally low dispersion of data about the regression line (`MSWD` = 0.15), clearly does not conform to this general trend.

Overall, we find that U–Pb isochron ages of similar average precision can be obtained throughout the ∼630–430 ka interval considered in this study provided that the isochron either encompasses data points with a reasonable spread in Pb/U ratios or includes highly radiogenic material. In practical terms, there are a number of ways in which samples can be pre-screened for suitability in this regard. A number of options were discussed in Woodhead et al. (2012) but advances in both chemical separation procedures (allowing rapid sample throughput) and the development of in situ analytical techniques (e.g. Roberts et al., 2020) have now rendered some of these approaches redundant. Currently, the most accurate and time-efficient methods for

assessing suitability for U–Pb dating would appear to be either pre-screening via laser ablation ICPMS (Woodhead and Petrus, 2019), U–Th isotopic analysis using a high-throughput protocol (e.g. Hellstrom, 2003) whereby $^{232}$Th is employed as a proxy for inherited Pb (Woodhead et al., 2006), or simple reconnaissance ID analysis involving a small number (∼3) of sub-samples taken from different parts of the growth domain.

### 4.3 Comparative of age precision


Given that U–Pb age uncertainties do not show an obvious trend with increasing age (Fig. 4), it is reasonable to characterise average expected U–Pb age uncertainties using either a mean (±10.6 ka, $2\sigma$) or median (±8.5 ka, $2\sigma$) value. In this context, however, we argue that the median provides a more useful indicator of typical uncertainty because it reduces the influence of outliers—specifically, samples CC2-1 and CC17-1-3, which exhibit anomalously high age uncertainties compared to the

other isochrons. For this dataset, median U–Pb age uncertainties exceed expected U–Th uncertainties at ∼520 ka, suggesting that U–Pb dating may generally yield higher precision ages beyond this point. However, caution is needed in generalising this finding to other study sites or sample types, which may be either less suitable to U–Pb geochronology (e.g. higher average inherited Pb) or more suitable to U–Th geochronology (e.g. similar U and Th content, but higher initial $\left[^{234}\mathrm{U}/^{238}\mathrm{U}\right]$). In many cases, it is expected that this crossover point would occur at a somewhat older age.

Furthermore, even in cases where U–Pb dating delivers higher precision for a individual ages, there may be practical reasons to favour use of the U–Th method. For example, U–Pb isochron ages are significantly more labour intensive on a per-age basis (when using precise ID MC-ICP-MS and TIMS approaches) compared to U–Th ages because of the need to analyse multiple coeval sub-samples (typically >4). Thus, in applications combining multiple age determination, such as in compiling a depth-age model along the axial length of a speleothem (e,g. Scholz et al., 2012; Bajo et al., 2012), the U–Th method may achieve

lower final uncertainties for a given amount of analytical effort, even beyond the precision crossover point.

### 4.4 Inherited $^{207}$Pb/$^{206}$Pb composition

In addition to yielding an age solution, Tera-Wasserburg isochrons yield an estimate of the $^{207}$Pb/$^{206}$Pb composition of inherited Pb, assuming this remains constant across the sub-samples used to construct the isochron (Tera and Wasserburg, 1972; Woodhead et al., 2012). The availability of multiple Tera-Wasserburg isochrons in this study, therefore, provides an opportu-

nity to assess the consistency of the isotopic composition of inherited Pb in these samples. We find that inherited $^{207}$Pb/$^{206}$Pb values are reasonably consistent (weighted mean = 0.8148 ± 0.0012, MSWD = 3.3, $n = 14$, $p$=0.00) with no apparent trend with

age, although the elevated `MSWD` value indicates the data are overdispersed with respect to their analytical uncertainties. If we assume that this overdispersion is due to real variability in the inherited Pb isotopic composition, and furthermore, that this additional component of variability is independent and follows a strict Gaussian distribution (e.g. Ludwig, 2000), we obtain a weighted average of $0.8149 \pm 0.0017$ ($\sigma_{excess} = 0.023$). However, the assumption of Gaussian-distributed excess scatter is not necessarily justified by the available data here. These results are reasonably close to the average $^{207}\text{Pb}/^{206}\text{Pb}$ value of Bajo et al. (2020), who obtained a weighted mean value of $0.8134 \pm 0.0048$ for Corchia Cave speleothems deposited between ~970–810 ka.

The relative consistency of inherited $^{207}\text{Pb}/^{206}\text{Pb}$ values observed in this study provides an opportunity to assess the effect of fixing Tera-Wasserburg isochron slopes to a common $y$-intercept value. If we repeat the U–Pb age calculations with the Tera-Wasserburg isochron slopes fixed at 0.8148, all ages remain within their original unanchored isochron age uncertainties, except for sample CCB-F-1, which returns a marginally older U–Pb age, and CC2-1, which returns a significantly older age. It is worth noting that isochron CC2-1 exhibits poor spread in U/Pb ratios and high average inherited Pb, and for these reasons it is particularly sensitive to the effect of anchoring the y-intercept. Overall, these results suggest that for isochron data sets with limited U/Pb variation, anchoring $y$-intercepts to a well-constrained common $^{207}\text{Pb}/^{206}\text{Pb}$ value may still produce reliable ages at Corchia Cave.

### 4.5  Speleothem $^{238}\text{U}/^{235}\text{U}$ values

In recent years, high-precision U-isotope analyses have revealed significant natural $\delta^{238}\text{U}$ variations in a wide range of materials. These variations are attributed primarily to U fractionation associated with nuclear volume effects rather than classical mass-dependent fractionation (Fujii et al., 2009) and appear to be most pronounced during redox reactions. Notably, during the reduction of soluble $\text{U}^{6+}$ to insoluble $\text{U}^{4+}$, $^{238}\text{U}$ is concentrated in the $\text{U}^{4+}$ state, leaving the insoluble phase with a higher $\delta^{238}\text{U}$ than the remaining dissolved U. This forms the basis for efforts to employ $\delta^{238}\text{U}$ of marine sediments as a palaeoredox proxy (e.g. Andersen et al., 2014). Processes such as adsorption (e.g. Weyer et al., 2008) and mineral leaching by strong acids (Stirling et al., 2007; Hiess et al., 2012) have also been observed to induce $^{238}\text{U}/^{235}\text{U}$ fractionation, while oxidation of U is not associated with significant fractionation.

Data acquired over the past ~15 years suggest that the average $\delta^{238}\text{U}$ of bulk crustal rocks (~-0.29 ‰) is practically indistinguishable from that of the deep mantle ($\sim -0.31$ ‰) (Andersen et al., 2017; Tissot and Dauphas, 2015). The upper mantle is somewhat more enriched in $^{238}\text{U}$ on average due to recycling of altered oceanic crust (Andersen et al., 2015), whereas modern ocean water and marine carbonates are more depleted in $^{238}\text{U}$ ($\sim -0.39$ ‰ Tissot and Dauphas, 2015). Accessory U minerals show more $\delta^{238}\text{U}$ variability than bulk crustal rocks, as do materials formed in low-temperature environments (Stirling et al., 2007; Weyer et al., 2008; Hiess et al., 2012). To date, only a small number of studies have reported $\delta^{238}\text{U}$ values for speleothems. Stirling et al. (2007) and Cheng et al. (2013) presented $\delta^{238}\text{U}$ values for speleothems from numerous cave systems and observed highly variable $\delta^{238}\text{U}$ values ranging from $\sim -0.7$ to +0.4 ‰, although variability within a single

speleothem or cave system was significantly lower (Shen et al., 2012). Stirling et al. (2007) also identified a tentative correlation between speleothem $\delta^{238}$U and measured $[^{234}\text{U}/^{238}\text{U}]$ values, suggesting that speleothem $^{234}$U-disequilibrium and $^{238}$U/$^{235}$U fractionation may be linked by a common weathering control, albeit involving distinct fractionation mechanisms.

The use of a high precision all-Faraday-cup protocol for U–Th measurements in this study, which employs a $^{233}$U-$^{236}$U double spike for mass bias correction, enabled accurate and precise $\delta^{238}$U determinations. Average analytical uncertainty in measured $^{238}$U/$^{235}$U ratios was 0.0006 ($2\sigma$), which is adequate for assessing $\delta^{238}$U variability at the sub-‰ level (see Table 1). A comparison of our speleothem $\delta^{238}$U with previous studies (Fig. 6) reveals that most values fall within a range from approximately $-0.6$ to $-0.2$ ‰, consistent with the range observed for carbonate rocks globally (Li and Tissot, 2023). This supports the simple hypothesis that speleothems primarily inherit their $^{238}$U/$^{235}$U composition from carbonate host rocks without significant fractionation. However, two Nullarbor speleothem samples (MO-1/3; Woodhead et al., 2006) and the KOZ samples (Stirling et al., 2007) exhibit elevated $\delta^{238}$U values, suggesting either a more $^{238}$U-enriched non-carbonate source of U, or the presence of an enriching fractionation process during source water U uptake and/or transport to the site of speleothem deposition. Clearly further studies, ideally involving paired speleothem-bedrock analyses, are required to investigate the cause of these high $\delta^{238}$U values, as well as the primary controls on speleothem $\delta^{238}$U values more generally.

We also compared the $\delta^{238}$U and measured $[^{234}\text{U}/^{238}\text{U}]$ values of the speleothem data obtained in this study with data from previous studies to further assess the relationship between speleothem $^{234}$U-disequilibrium and $^{238}$U/$^{235}$U fractionation (see Fig. S5 in The Supplement). With the inclusion of additional speleothem data from this study and others, we observed a weaker correlation between $\delta^{238}$U and measured $[^{234}\text{U}/^{238}\text{U}]$ values than Stirling et al (2007), i.e. $R^2 = 0.42$ versus $R^2 = 0.65$, and this correlation becomes negligible if the 'high leverage' KOZ data are excluded. However, for speleothem samples that are presumed to have acted as a closed system with respect to U-series isotopes post deposition, it is arguably more appropriate to compare $\delta^{238}$U values with initial $[^{234}\text{U}/^{238}\text{U}]$ values instead of measured activity ratios, since measured $[^{234}\text{U}/^{238}\text{U}]$ values also depend on sample age. When repeating this comparison using initial $[^{234}\text{U}/^{238}\text{U}]$ values instead of measured values (necessarily excluding a number of samples for which no age data was available), no correlation is observed ($R^2 = 0.01$). This result is inconsistent with the hypothesis that speleothem $^{234}$U-disequilibrium and $^{238}$U/$^{235}$U fractionation are linked by a common weathering control.

Regardless of the dominant controls on speleothem $\delta^{238}$U values, the data available thus far show that the U isotopic composition of most speleothems departs significantly from the conventional $^{238}$U/$^{235}$U value of Steiger and Jäger (1977), and also to a lesser extent, the average terrestrial zircon value of Hiess et al. (2012). Using a present-day $^{238}$U/$^{235}$U value of $\sim$137.79 (the weighted average obtained for speleothem samples in this study) in place of the traditional value of 137.88 (Steiger and Jäger, 1977), either for mass bias correction of U–Pb data and/or in U–Pb age calculation, has a negligible effect on U–Pb ages for young (e.g. Cenozoic) samples. However, when applying the Pb/Pb dating approach to older materials, the $^{238}$U/$^{235}$U value adopted can have a significant effect on age accuracy (Stirling et al., 2007; Weyer et al., 2008; Hiess et al., 2012). This is also the case in U–Th dating using an assumed $^{238}$U/$^{235}$U value either for mass bias correction or calculating $^{238}$U-based ratios

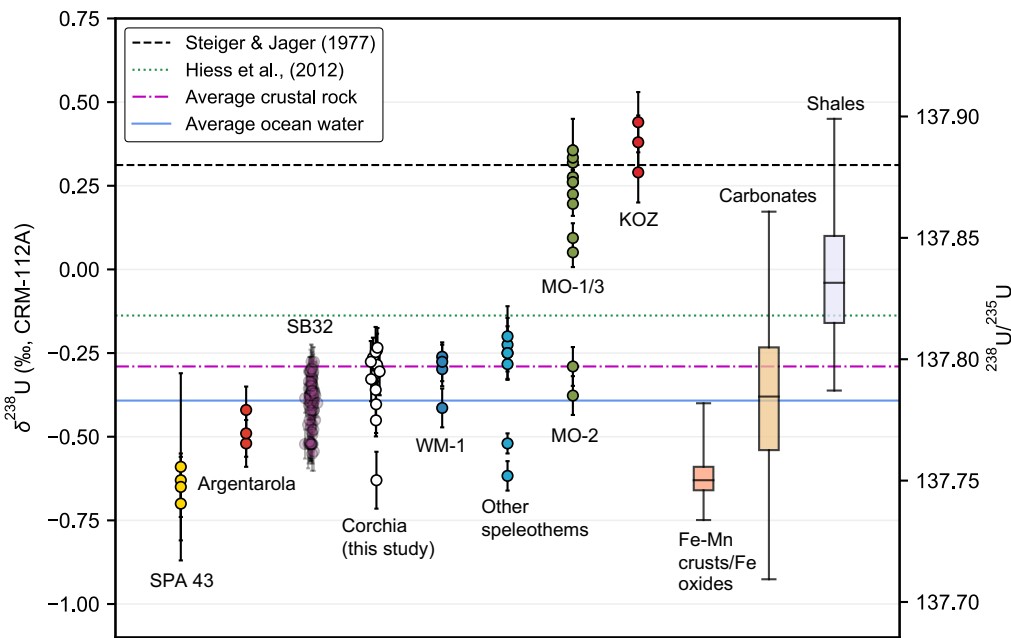

**Figure 6.** Speleothem $\delta^{238}U$ data from this study and others. Speleothem samples are grouped according to cave site where more than two analyses were available. Speleothem data from Cheng et al., (2013) include SB32 from Sanbao Cave, central China (e.g. Wang et al., 2008), WM1 from Wilder Mann Cave, Northern Calcareous Alps, Austria (Meyer et al., 2009), and MO-1/2/3 from Leana's Breath Cave, Nullarbor Plain, Australia (Woodhead et al., 2006). 'Other speleothems' include the GBW04412 and GBW04413 stalagmite standards from Wang et al. (2023), the FAB-LIG Sample from Stirling et al., (2007), and the Kr3 and CA-1 samples from Cheng et al., (2013). Speleothem samples analysed by Stirling et al. (2007) include SPA-43 from Spannegal Cave, Austria (e.g. Spötl and Mangini, 2010), Argentaorola from Argentarola Island, central Italy (e.g. Bard et al., 2002), and KOZ. Also shown are the 'consensus' $^{238}U/^{235}U$ value of Steiger and Jäger (1977), the average terrestrial zircon value of Hiess et al., (2012), and the average modern ocean water (–0.39 ‰) and crustal rock (–0.29 ‰) values (Tissot and Dauphas, 2015). The global range of $\delta^{238}U$ values for carbonates, Phanerozoic shales, and Fe-Mn crusts/Fe oxides are shown as box plots with 'whiskers' indicating the 2.5 and 97.5 percentiles. These data were taken from the Uranium Isotope Database (UID) of Li and Tissot (2023). Absolute $^{238}U/^{235}U$ values are shown on the right-hand y-axis based on the CRM-112A $^{238}U/^{235}U$ value of 137.837 from Richter et al. (2010). Richter et al. (2010).

from measured $^{235}U$ (Stirling et al. 2007, Cheng et al. 2013), although the magnitude of the inaccuracy depends greatly on the specific analytical protocol that is adopted (Shen et al., 2012). The significant departure of speleothem $^{238}U/^{235}U$ values from 137.88 that is observed here re-iterates the need to pay careful attention to the $^{238}U/^{235}U$ value used when dating carbonates under these circumstances.


# 5  Conclusions

This comparison of the carbonate U–Th and U–Pb chronometers demonstrates excellent agreement between the two methods over a substantial part of the Middle Pleistocene (∼630–430 ka). In addition to establishing consistency, we identify key differences in their performance characteristics. We find that:


1. U–Th age uncertainties are reasonably predictable (for a given level of analytical precision) and tend to increase in an approximately exponential manner with age, whereas U–Pb age uncertainties do not increase systematically over this interval but are highly dependent on individual sample characteristics (U/Pb spread of data points and/or the availability of highly radiogenic material).

2. U–Pb isochron age precision surpasses that of U–Th dating prior to the latter reaching its upper age limit. This occurs
at ∼520 ka in our dataset, although the exact crossover point is expected to vary accross different sample types and depositional settings.

3. Even in cases where the precision of U–Pb isochron ages clearly surpasses that of a corresponding U–Th age determinations, U–Th dating may still offer some practical advantages. For example, when combining multiple age determinations (e.g. in compiling speleothem depth-age models), because it is generally less labour intensive on a per-age basis.

Overall, these findings highlight the complementary nature of the U–Th and U–Pb dating methods, and reaffirm the potential for their combined use in producing accurate and consistent Middle Pleistocene chronologies.

Our assessment of speleothem $^{238}U/^{235}U$ reveals that:

1. The majority of speleothem $^{238}U/^{235}U$ values are generally consistent with the range observed for carbonate rocks globally, although there are some exceptions.

2. Speleothem $^{238}U/^{235}U$ values typically deviate significantly from the nominal value of 137.88 (Steiger and Jäger, 1977), traditionally adopted as a constant in geochronology. This re-emphasises the need to pay careful attention to the value adopted in data processing and age calculation in cases where it has a significant influence on calculated ages.

3. Further studies are required, ideally involving paired speleothem-bedrock analyses, to better understand the controls on speleothem $^{238}U/^{235}U$.

*Data availability.*  All data used for age calculations and subsequent statistical analyses are available in The Supplement.

## Appendix A: Age equations

### A1 U–Th age equation

The standard U–Th age equation, which assumes negligible initial $^{230}$Th, may be written as

$$A_{08} = 1 - e^{-\lambda_{230}t} + \frac{\lambda_{230}}{\lambda_{230} - \lambda_{234}} (A_{48} - 1) \left(1 - e^{(\lambda_{234} - \lambda_{230})t}\right) \tag{A1}$$

where $A_{08}$ and $A_{48}$ denote measured $\left[^{230}\text{Th}/^{238}\text{U}\right]$ and $\left[^{234}\text{U}/^{238}\text{U}\right]$ activity ratios (Broecker, 1963). This equation can be solved for $t$ via iterative methods, such as Newton's method.

### A2 U–Pb age equation

The Tera-Wasserburg U-Pb age equation may be written as

$$N_{75} = U \left(a N_{68} + b\right) \tag{A2}$$

where $N_{75} = \frac{^{207}\text{Pb}^*}{^{235}\text{U}}$ (such that the superscript * denotes radiogenic Pb), $N_{68} = \frac{^{206}\text{Pb}^*}{^{238}\text{U}}$, U is the $\frac{^{238}\text{U}}{^{235}\text{U}}$ ratio, and $a$ and $b$ are the isochron y-intercept and slope respectively. If initial abundances of intermediate nuclides other than $^{234}$U are negligible, then Equations 1 and 7 in Pollard et al., (2023) simplify to

$$N_{68} = \gamma + \frac{\lambda_{238}}{\lambda_{234}} \left((A_{48} - 1) e^{\lambda_{234}t} + 1\right) \eta \tag{A3}$$

and

$$N_{75} = e^{\lambda_{235}t} \left(d_1 e^{-\lambda_{235}*t} + d_2 e^{-\lambda_{231}t} + 1\right) \tag{A4}$$

where

$$\gamma = c_1 e^{-\lambda_{238}t} + c_2 e^{(\lambda_{238} - \lambda_{234})t} + c_3 e^{(\lambda_{238} - \lambda_{230})t} + c_4 e^{(\lambda_{238} - \lambda_{226})t} + 1$$

$$\eta = h_1 e^{(\lambda_{238} - \lambda_{234})t} + h_2 e^{(\lambda_{238} - \lambda_{230})t} + h_3 e^{(\lambda_{238} - \lambda_{226})t} + e^{\lambda_{238}t}$$

and $c_i$, $h_i$ and $d_i$ are Bateman coefficients given by

$$c_1 = \frac{-\lambda_{234}\lambda_{230}\lambda_{226}}{(\lambda_{234} - \lambda_{238})(\lambda_{230} - \lambda_{238})(\lambda_{226} - \lambda_{238})}$$

$$c_2 = \frac{-\lambda_{234}\lambda_{230}\lambda_{226}}{(\lambda_{238} - \lambda_{234})(\lambda_{230} - \lambda_{234})(\lambda_{226} - \lambda_{234})}$$

$$c_3 = \frac{-\lambda_{234}\lambda_{230}\lambda_{226}}{(\lambda_{238} - \lambda_{230})(\lambda_{234} - \lambda_{230})(\lambda_{226} - \lambda_{230})}$$

$$c_4 = \frac{-\lambda_{234}\lambda_{230}\lambda_{226}}{(\lambda_{238} - \lambda_{226})(\lambda_{234} - \lambda_{226})(\lambda_{230} - \lambda_{226})}$$

$$h_1 = \frac{-\lambda_{230}\lambda_{226}}{(\lambda_{230} - \lambda_{234})(\lambda_{226} - \lambda_{234})}$$

$$h_2 = \frac{-\lambda_{234}\lambda_{226}}{(\lambda_{234} - \lambda_{230})(\lambda_{226} - \lambda_{230})}$$

$$h_3 = \frac{-\lambda_{234}\lambda_{230}}{(\lambda_{234} - \lambda_{226})(\lambda_{230} - \lambda_{226})}$$

$$d_1 = \frac{-\lambda_{231}}{(\lambda_{231} - \lambda_{235})}$$

$$d_2 = \frac{-\lambda_{235}}{(\lambda_{235} - \lambda_{231})}$$

Having fitted a suitable isochron regression line and obtained a $\left[^{234}\text{U}/^{238}\text{U}\right]$ measurement that is representative of the sampled material, Eq. A2 may also be solved for t via iterative methods, such as Newton's method.

## Appendix B: Age uncertainties and correlation

U–Th and U–Pb age uncertainties, as well as the uncertainty correlation coefficient, may be computed via first-order uncertainty propagation. A convenient way to set up these calculations is via the matrix approach outlined in McLean et al. (2011). Starting with Eq. 63 in Mclean et al. (2011) we have

$$\begin{bmatrix} \sigma^2_{t_{Th}} & \text{cov}(t_{Th}, t_{Pb}) \\ \text{cov}(t_{Th}, t_{Pb}) & \sigma^2_{t_{Pb}} \end{bmatrix} = \mathbf{J}^\top \cdot \mathbf{V} \cdot \mathbf{J} \tag{B1}$$

where $\mathbf{J}$ is the Jacobian matrix of derivatives and $\mathbf{V}$ is the covariance matrix of the age equation variables. These matrices may be written in full as

$$\mathbf{J} = \begin{bmatrix} \frac{dt_{\text{Th}}}{dA_{48}} & \frac{dt_{\text{Pb}}}{dA_{48}} \\ \frac{dt_{\text{Th}}}{dA_{08}} & 0 \\ 0 & \frac{dt_{\text{Pb}}}{da} \\ 0 & \frac{dt_{\text{Pb}}}{db} \end{bmatrix} \tag{B2}$$

$$\mathbf{V} = \begin{bmatrix} \sigma_{A_{48}}^2 & 0 & 0 & 0 \\ 0 & \sigma_{A_{08}}^2 & 0 & 0 \\ 0 & 0 & \sigma_a^2 & \text{cov}(a,b) \\ 0 & 0 & \text{cov}(a,b) & \sigma_b^2 \end{bmatrix} \tag{B3}$$

The derivatives in the first column of $\mathbf{J}$ are given by

$$\frac{dt_{\text{Th}}}{dA_{48}} = -\frac{1}{k_1} \left( \frac{\lambda_{230}}{\lambda_{230} - \lambda_{234}} \right) \left( 1 - e^{(\lambda_{234} - \lambda_{230})t} \right) \tag{B4}$$

and

$$\frac{dt_{\text{Th}}}{dA_{08}} = \frac{1}{k_1} \tag{B5}$$

where

$$k_1 = \lambda_{230} \left( e^{-\lambda_{230}t} + (A_{48} - 1) e^{(\lambda_{234} - \lambda_{230})t} \right) \tag{B6}$$

These expressions are equivalent to those given in Ludwig and Titterington (1994). The derivatives in the second column of $\mathbf{J}$ are given by

$$\frac{dt_{Pb}}{dA_{48}} = -\frac{1}{k_2} \left( \frac{\lambda_{238}}{\lambda_{234}} \right) a\eta e^{\lambda_{234}t} \tag{B7}$$

$$\frac{dt_{Pb}}{da} = -\frac{N_{68}}{k_2} \tag{B8}$$

$$\frac{dt_{Pb}}{db} = -\frac{1}{k_2} \tag{B9}$$

where

$$k_2 = a\frac{\partial N_{68}}{\partial t} - \frac{1}{U}\frac{\partial N_{75}}{\partial t} \tag{B10}$$

The partial derivatives in $k_2$ are then

$$\frac{\partial N_{68}}{\partial t} = \frac{\partial \gamma}{\partial t} + \frac{\lambda_{238}}{\lambda_{234}} \left( \frac{\partial \eta}{\partial t} \left( (A_{48} - 1)e^{\lambda_{234}t} + 1 \right) + \eta(A_{48} - 1)\lambda_{234}e^{\lambda_{234}t} \right)$$

where $\frac{\partial \gamma}{\partial t}$ and $\frac{\partial \eta}{\partial t}$ are trivial to compute since $\gamma$ and $\eta$ are simply composed of sums of exponential terms, and

$$\frac{\partial N_{75}}{\partial t} = (\lambda_{235} - \lambda_{231})d_2 e^{(\lambda_{235} - \lambda_{231})t} + \lambda_{235}e^{\lambda_{235}t}$$

Having calculated the U–Th and U–Pb age uncertainties ($\sigma_{t_{Th}}$, $\sigma_{t_{Pb}}$) and the covariance between them (i.e. $\text{cov}(t_{Th}, t_{Pb})$) via Eq. A5, the correlation coefficient is simply given by

$$\rho = \frac{\text{cov}(t_{Th}, t_{Pb})}{\sigma_{t_{Th}}\sigma_{t_{Pb}}} \tag{B11}$$

*Author contributions.* GZ, RD, II, JH, ER, and TP collected the samples, and undertook preliminary analyses to identify those suitable for this study. RLE, HC, MP, and XL, developed the U–Th analytical protocol, and TP, MP, XL, and DP carried out the U–Th analyses. TP, JW, and AW carried out the U–Pb analyses. TP undertook the statistical analyses and wrote the manuscript with contributions from all co-authors.

*Competing interests.* The authors declare that they have no competing interests.

*Acknowledgements.* We are grateful to the Gruppo Speleologico Lucchese for their assistance with the recovery of speleothem samples used in this study. We thank Cameron Patrick of the Statistical Consulting Centre, University of Melbourne, for advice on certain aspects of the age comparison. This research was funded by Australian Research Council Discovery Project grants DP160202969 (to RD, JW, JH, ER and GZ), DP220102133 (to RD, JW, and ER), and FL160100028 to JW. We thank David Richards and one anonymous reviewer for their helpful comments and suggestions.

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
