# Peer review of "Radiometric dating of Middle Pleistocene carbonates: assessing consistency and performance of the U–Th and U–Pb dating methods"

_EGUsphere, 2024_

## Author Comment (AC1)

**Response to RC2 on 'Double dating in the Middle Pleistocene: assessing the consistency and performance of the carbonate U–Th and U–Pb dating methods'**

We thank the reviewer for their careful reading of the manuscript and helpful suggestions. We respond to the reviewer's comments below.

**1. Specific comments**

Line 6: thousand of years (ka)

We have changed this in the revised manuscript.

Line 28: applications

This seems correct without the 's'?

Line 48: worth mentioning briefly why U–Pb chronometer is currently less widely applied than U-Th for this time interval.

We have added a sentence to this paragraph explaining that the U–Pb method is less widely applied to Pleistocene carbonates than the U–Th method mostly because it is more labour intensive and only yields precise ages for samples with low inherited Pb.

What does UoM stands for, I assume University of Melbourne?

Does this comment relate to the caption of Figure 1? UoM does stand for University of Melbourne. We have now specified this in the caption of Figure 1.

Line 245: The U–Pb age of sample CCB-6-1 seems to be reliable based on good spread in U/Pb ratios along the isochron and data point scatter. However, the discussion on the synchronicity of the stable isotope profiles among the 3 stalagmites lacks sufficient context. The authors should elaborate on the basis of this agreement, and provide some more context about the good agreement between isotopic variations amongst coeval speleothems from Galleria delle Stalattiti.

We have largely addressed this in our response to reviewer 1.

We have modified this sentence to add more context. The good agreement between isotopic profiles of coeval GdS speleothems is demonstrated in Tzedaksi et al. (2018) and Bajo et al. (2020). We have added the Tzedakis et al. (2018) citation to the revised text.

Line 260-263: the MSWD and p values listed here for the best-fit regression line and for goodness-of-fit of the paired ages to a 1:1 line are different than the values listed in Figure 3. Please revisit either the text of the figure to reflect the correct values.

In the text we provide two different MSWD/p-values. The first (line 260) is for an unconstrained regression fit to the data. We then check if the slope is consistent with one within uncertainties, and the y-intercept is consistent with zero. The second (line 262) pertains to a fit of the data to a fixed 1:1 line passing through the origin. Since we plot the data in Figure 3 against a 1:1 line, this is the MSWD value provided in the text box.

Line 274: I think the text here refers to figure 4, not 5. Please correct.

We have corrected this.

Line 313: specify for the reader that this Figure S4 in the Supplemental Material

Done.

Line 381: include here the references of Woodhead et al (2006) and Stirling et al (2007) for the speleothem MO-1/3 and KOZ.

We have added these in.

Line 387: see Supplemental figure S5

Done.

Line 388: worth mentioning what is the value for this correlation in Stirling et al (2007).

We have added in this value.

**2. Figures and tables**

Table 1:   note in the table legend that CI stands for confidence interval

Done.

Figure 1: it is difficult to see the grey shading indicating the 95% confidence interval of the black line; consider using a darker shade.

We have adjusted this accordingly.

Figure 2:   in the legend, specify what MSWD stands for and that n is for number of samples.

We have added this information.

Table 2: this table is not referenced in the manuscript.

We have added a reference to this table in the text.

In the table legend, what does "3 Assuming Gaussian distributed analytical uncertainties" refers to? Number 3 is not listed in the table.

The superscript of the heading of the second column should have read '2, 3'. This was accidentally omitted.

Figure 6: For consistency in the figure legend and enhance readability, I suggest including the following information:

" consensus value in geochronology"( Steiger & Jager (1977)

average terrestrial zircon (Hiess et al (2012)

average crustal rock (Tissot and Dauphas) 2015

average oven water (Tissot and Dauphas) 2015

While we agree that it would be convenient to have this information available in the legend, doing so makes the plot overly cluttered. We have provided this information in the caption and hope that this will suffice for most readers.

In the figure caption replace "Li et al" with "Li and Tissot"

We have corrected this.

**References**

Bajo, P., Drysdale, R. N., Woodhead, J. D., Hellstrom, J. C., Hodell, D., Ferretti, P., Voelker, A. H. L., Zanchetta, G., Rodrigues, T., Wolff, E., Tyler, J., Frisia, S., Spötl, C., & Fallick, A. E. (2020). Persistent influence of obliquity on ice age terminations since the Middle Pleistocene transition. *Science*, *367*(6483), 1235–1239. https://doi.org/10.1126/science.aaw1114

Tzedakis, P. C., Drysdale, R. N., Margari, V., Skinner, L. C., Menviel, L., Rhodes, R. H., Taschetto, A. S., Hodell, D. A., Crowhurst, S. J., Hellstrom, J. C., Fallick, A. E., Grimalt, J. O., McManus, J. F., Martrat, B., Mokeddem, Z., Parrenin, F., Regattieri, E., Roe, K., & Zanchetta, G. (2018). Enhanced climate instability in the North Atlantic and southern Europe during the Last Interglacial. *Nature Communications*, *9*(1), 1383–14. https://doi.org/10.1038/s41467-018-06683-3

---

## Author Comment (AC2)

**Response to RC1 on 'Double dating in the Middle Pleistocene: assessing the consistency and performance of the carbonate U–Th and U–Pb dating methods'**

We thank David Richards for his careful reading of the manuscript and thorough review. We respond to the reviewer's comments below.

**1. Specific comments**

**1.1. On the term 'double dating'**

Comment: I accept that this term is becoming more widely used, but please acknowledge that U–Th and U–Pb methods are not independent – in time we will be measuring all isotopes in the decay chain to assess the state of disequilibrium from U to Pb! The key here is that both U–Th and U–Pb protocols are being applied to sub-samples of calcite formed at the same time.

We have removed the term 'double dating' from the title so as to avoid any confusion. We have also added a sentence to the introduction explicitly stating that the U-Th and U–Pb dating methods are not entirely independent because they both rely on the initial part of the $^{238}$U decay series, and that this necessarily limits our ability to verify the absolute accuracy of these dating methods via a direct age comparison.

Comment: It would be appropriate to signpost similar strategies of combined or ?double-dating approaches from recent and deeper past (e.g.  U, Th – He vs U/Th; U–Pb vs U–Th-Sm/He). You might even include ESR – U–Th comparison studies (see some refs below).

Now that we have modified the title we believe that a review of other studies comparing different double dating methods is outside the scope of this manuscript, but have added a sentence acknowledging that other methods of dating carbonates within this age range are available.

**1.2 Age consistency?**

Comment: Is it justifiable to consider these as independent ages and use a simple Z-test to assess the statistical difference? I don't know what the solution is here, but much of the uncertainty is shared (e.g. measured $^{234}$U/$^{238}$U).

As we attempted to convey in the text, we do not assume that the U–Th and U–Pb ages for each sample are statistically independent, but rather account for the correlation that arises as a result of the shared $^{234}$U/$^{238}$U measurement uncertainty. We do this as follows:

For each sample, we compute the age difference as

$$\Delta = t_{Th} - t_{Pb}$$

Uncertainty on the age difference is then be computed via first-order error propagation as

$$\sigma_\Delta = \sqrt{\sigma_{t_{Th}}^2 + \sigma_{t_{Pb}}^2 - 2\rho\sigma_{t_{Th}}\sigma_{t_{Pb}}}$$

where $\sigma_\Delta$ is the standard error on the age difference, $\sigma_{t_{Th}}$ and $\sigma_{t_{Pb}}$ are the standard errors on the U–Th and U–Pb ages respectively, and $\rho$ is the correlation coefficient, which is non-zero due to the shared $^{234}U/^{238}U$ measurement uncertainty (more on calculating $\rho$ below).

The test statistic is then given by

$$z = \frac{\Delta}{\sigma_\Delta}$$

If $|z|$ is greater than 1.96 then the ages do not agree at the $\alpha = 0.05$ significance level. A formal $p$-value may be computed by comparing $z$ against the standard normal distribution. As stated in the manuscript, an assumption implicit in the above procedure is that the age difference (i.e. $\Delta$) PDF is Gaussian distributed, and this is not strictly applicable for some of the older samples where the U–Th age uncertainty distribution is slightly skewed. Therefore, for these samples, we use a Monte Carlo procedure to assess age consistency in place of the formal hypothesis test.

**Table 1**: Comparison U–Th versus U–Pb age uncertainty correlation coefficients calculated using an algebraic approach and the Monte Carlo approach with $10^6$ iterations.

| Sample ID | $\rho$ | $\rho$ (Monte Carlo) |
|---|---|---|
| CCB-B-1 | 0.112 | 0.113 |
| CCB-C-3 | 0.030 | 0.029 |
| CCB-C-1 | 0.015 | 0.016 |
| CCB-C-20 | 0.023 | 0.023 |
| CC17-1-1 | 0.041 | 0.041 |
| CC17-1-35 | 0.032 | 0.033 |
| CCB-E-9 | 0.032 | 0.034 |
| CCB-E-10 | 0.037 | 0.038 |
| CCB-F-16 | 0.058 | 0.059 |
| CCB-F-1 | 0.059 | 0.058 |
| CC2-1 | 0.008 | 0.009 |
| CCB-6-1 | 0.096 | 0.096 |
| CC17-1-3 | 0.028 | 0.028 |
| CC15-1 | 0.111 | 0.111 |

In the original submission we computed $\rho$ using a Monte Carlo simulation and found that these values, i.e. the age uncertainty correlation coefficients, are actually relatively small, indicating that the effect of shared [$^{234}$U/$^{238}$U] uncertainty is minor. However, it is also possible to compute $\rho$ algebraically instead. We do this in the revised manuscript for improved clarity (adding full details of the calculations) but note that the difference between the $\rho$ values calculated algebraically and by Monte Carlo simulation is negligible (see Table 1 below).

Comment: You draw upon additional data (stable isotope variation) to demonstrate the accuracy (or consistency) between U–Pb and U–Th ages – i.e. tie-points and stratigraphic position. Is this data to appear in a future publication? Is there comparison with the timing of Milankovitch forcing involved? One must take the success of this strategy at face value because the data are not illustrated.

We don't actually use the stable isotope/tie-point data to assess consistency between the ages: this is done solely on the basis of sampling material from the same stratigraphic positions and then assessing how well the two ages agree.

It is true that we have generated $\delta^{18}$O and $\delta^{13}$C profiles for these stalagmites, and it is our intention to publish these data in future. Some of these data will be included in the first author's PhD thesis. We don't include these data in the current manuscript because this dataset is still a work in progress and outside the scope of the present study.

We only reference these data in relation to sample CCB-6-1, after having already concluded that the two ages don't agree based on a direct comparison. In this case, we note that the U–Pb age is more consistent with the other ages when we consider stratigraphic position of all the samples. Doing so requires transferring the U–Th /U–Pb sampling positions to a common depth scale by synchronising the isotope profiles of the three stalagmites using tie points. We are able to do this because stalagmites from the Galleria della Stallactiti chamber of Corchia Cave tend to display excellent replicability of their stable isotope profiles, as demonstrated by Tzedakis et al. (2018) and Bajo et al. (2020). We have amended the text here slightly to clarify these points.

It is important to note that the use of these additional isotope/tie-point data is not central to any of the main findings of this study.

**1.3 Age precision**

Comment: Section 4.2. The nature of **contribution** to age uncertainty differs considerably between the two chronometers. This is the crux of the paper, but at times the text needs reworking. You describe here a specific case with low initial Th, were this to be more significant, there would be more similarity between the sources of uncertainty.

We focus on samples with low initial Th because, for the most part, the only Middle

Pleistocene carbonates that are suitable for U–Pb dating are those with low inherited Pb. Given that Th and Pb exhibit similar chemical behaviour in typical speleothem and coral forming environments, such samples will almost always have low initial Th. In fact, a large number of carbonate U–Th and U–Pb analyses conducted over the years at the University of Melbourne suggest that practically all samples with significant initial Th also have high inherited Pb (although the converse isn't necessarily true, i.e. it appears possible to occasionally have high inherited Pb without high initial Th, suggesting multiple pathways of Pb incorporation).

In our view, it is not obvious that there will be a significant correlation in the main sources of uncertainty as the amount of initial Th increases. This is because correction for initial Th has a smaller relative effect on older U–Th ages, and at the point where uncertainties associated with initial Th correction become significant relative to other sources of uncertainty, it is likely that U–Pb isochron ages will be highly degraded due to the amount of inherited Pb present.

We have added a sentence to the introduction stating that the applicability of U–Pb dating to Pleistocene carbonates is limited to samples with low inherited Pb. We have also re-worked the text in this section, and state that our findings only apply to carbonates with low initial Th/inherited Pb.

Comment: Use of the term 'predictable uncertainties' is awkward. There are fewer 'degrees of freedom' in your U–Th Corchia case, where initial Th is minimal and uncertainties are dictated by solutions of the age equation towards its upper limit. Your work is exploratory but does not offer definitive rules or solutions.

As discussed above, we focus on low initial Th carbonates in this study. For samples with low initial Th, U–Th age uncertainties are reasonably predictable for a given level of analytical precision. We don't claim to offer any definitive rules or solutions in this regard.

Comment: There is no assessment of the variation in initial $^{207}Pb/^{206}Pb$... or reported data. It would be useful to make a comment here. I presume the age calculations on not based on anchored common Pb.

The U–Pb isochron ages presented in the manuscript are not based on anchoring the y-intercept to a common initial Pb value.

Estimates of initial $^{207}Pb/^{206}Pb$ values obtained from the Tera-Wasserburg isochron fits are reasonably consistent across the samples. Using a classical statistics algorithm, we obtain a weighted average $^{207}Pb/^{206}Pb_i$ value of $0.8148 \pm 0.0012$ and a MSWD of 3.3 ($n=14$, $p=0.00$) for all U–Pb isochrons in this study. This result indicates that the data are overdispersed with respect to their analytical uncertainties. If we assume that this overdispersion is due to real variability in the inherited Pb of these speleothems, and that this variability is Gaussian i.i.d (e.g. Ludwig, 2000), we obtain a weighted average value of $0.8149 \pm 0.0017$ ($\sigma_{excess} = 0.023$).

It should be noted, however, that the assumption of Gaussian distributed excess scatter is not necessarily justified by the available data. These results are in good agreement with the average $^{207}Pb/^{206}Pb_i$ value obtained by Bajo et al. (2020) and other unpublished Corchia data.

We have added a brief discussion of these results to the revised manuscript.

**2. Response to technical comments**

Comment: Line 26: You refer to radiometric tools. Add information here on other tools (astro, geomagnetic and others).

We have restructured the introduction and no longer refer to 'radiometric tools'. As stated above, we have added a sentence acknowledging that other methods of dating carbonates within this age range are available.

Comment: Line 27. Utility determined by the material… vague – expand.

In restructuring the introduction we have removed this sentence.

Comment: Line 34: "Middle Pleistocene (ca. 400-650 ka)." Please consider the official term and dates associated with GSSP etc. see ref Head (2021) and elsewhere. Maybe you could refer to Chibanian (Middle Pleistocene).

This phrase reads '…portion of the Middle Pleistocene…'. It wasn't our intention to imply that the Middle Pleistocene as a whole spans 400–650 ka. We have rephrased this part of the text to avoid confusion.

Comment: Line 44: Is it better to declare that $^{230}Th$ **activity** approaches secular equilibrium with parent nuclides?

Agreed. We have changed this.

Comment: Line 45: The U–Th technique does not **impose** a limit of 650 ka, this is determined by the measurable extent of disequilibrium in the uranium-series decay chain.

We agree and have rephrased this sentence to make it more accurate.

Comment: Line 49: delete 'currently'

Done.

Comment: Line 53: Consider using the following… 'suited to dating older material, for which sufficient time has passed for significant accumulation of radiogenic Pb, it is also suitable for middle Pleistocene material that has ($^{234}U/^{238}U$) activity ratios that are …'

We appreciate the suggestion and have changed this sentence accordingly.

Is this meant to read line 56? If so, we stand by use of the term 'precise', although it should read '~270 ka' rather than '~200 ka'. Cliff et al. (2010) obtain a U–Pb age of 267 ± 1 ka for the youngest growth segment, which we consider to be reasonably precise. Admittedly the [U] of this flowstone (SPA4) is >100 ppm which is exceptionally high for a speleothem.

Done.

Fixed

We appreciated the suggestion and have changed this.

Line 136: 'sub-sample taken from the centre of the isochron' – rephrase.

We have rephrased this.

Line 141: Consider use of 'U and Th isotopic analysis was performed'

Changed.

Line 143: You declare a range of sample masses, but constant U abundance. Had you pre-screened the material for [U]?

The high-precision U–Th analysis was conducted after having acquired at least some of the U–Pb data. Therefore, we were able to use the [U] values from the U–Pb analyses to adjust the sample sizes and keep [U] relatively consistent for the U–Th analysis.

Also, prior to undertaking U–Pb analysis the stalagmites were pre-screened using a high-throughput U–Th procedure to obtain an approximate age, [U], and assess initial $^{230}$Th content, which we employ as a proxy for inherited Pb (e.g. Woodhead et al., 2006).

Line 153: Where were U and Th isotopic analyses conducted? MN, Melbourne,

The U–Pb analyses were conducted at The University of Melbourne and the U–Th analyses

were conducted at the University of Minnesota. We have added these details to the revised manuscript.

Line 160: Is it the lower dynamic range or the operational upper limit of applied current that precludes use of standard gain calibration methods.

The Neptune Plus MC-ICP-MS instrument that was employed to conduct the U–Th analyses uses a standard current of 3.33333 V (through a $10^{11}$ $\Omega$ resistor) in its automatic gain calibration procedure (see e.g. Wieser & Schwieters, 2005). This is suitable for $10^{11}$ and $10^{10}$ $\Omega$ resistors but surpasses the upper dynamic range of 0.5 V on the $10^{13}$ $\Omega$ resistor.

We have changed this (on line 163).

Line 176. Please clarify this statement 'corrected U ratios were normalised to CRM-112A'.

Samples measurements were bracketed by measurement of the CRM-112A standard. Subsequently, measured $^{234}U/^{238}U$ ratios were normalised to the value of $52.852 \pm 0.015$, obtained by Cheng et al. (2013). We have expanded on this in the revised manuscript.

Line 179. Please clarify (or expand upon) the following 'For the purposes of comparison... a correction for initial $^{230}Th$ was not applied'.

We have expanded upon this issue in the revised text.

For young samples with minor to moderate initial Th, It is of course possible to apply a correction for initial $^{230}Th$ to U–Pb ages using an equivalent approach to that adopted in U–Th geochronology (e.g. Cheng et al., 2000; Hellstrom, 2006). For example, initial $[^{230}Th/^{238}U]$ activity ratios could be estimated as

$$\left[\frac{^{230}Th}{^{238}U}\right]_i \approx \left[\frac{^{230}Th}{^{232}Th}\right]_i \left[\frac{^{232}Th}{^{238}U}\right]$$

where subscript i denotes an initial activity ratio, $[^{232}Th/^{238}U]$ is a measured activity ratio, and $[^{230}Th/^{232}Th]_i$ is estimated *a priori* based on an average bulk earth value (e.g. ~0.82) or a speleothem specific average value (e.g. ~1.5; Hellstrom, 2006). The estimated initial $[^{230}Th/^{238}U]$ activity ratio could then be inputted into Eq. 4 of Pollard et al. (2023) to account for the effect of initial $^{230}Th$ on $^{206}Pb$ accumulation. However, we believe that this approach is of limited utility because Pleistocene carbonates with significant initial Th are unlikely to be amenable to U–Pb dating due to the presence of high inherited Pb (as discussed above in section 1.3). In this study, a correction for initial Th results in an age correction of at most a few years for both U–Th and U–Pb ages. Therefore, for simplicity, we have opted to use that

standard U–Th and U–Pb age equations that do not include a correction for initial $^{230}$Th.

Line 194: On the use of $^{235}$U/$^{238}$U as internal mass bias. What value of $^{235}$U/$^{238}$U are you using? Does this come from your U–Th analysis?

Technically speaking it would make sense to use the $^{238}$U/$^{235}$U values obtained from our U–Th analyses in U–Pb data reduction and for Tera-Wasserburg isochron age calculation. However, for samples in this age range the difference between using the conventional $^{238}$U/$^{235}$U value of 137.88 and the values taken from our U–Th analysis (average ~137.79) is negligible. Therefore, for simplicity, we have opted to use the conventional value.

Line 202:    What do you mean by 'the data itself'? The rationale is probably embedded in Powell et al (2020) and Pollard et al (2023). Can you come up with a more useful phase here?

Generally speaking, there are two approaches to accounting for data scatter in fitting a regression line. The first, and probably most common in isochron dating, is to assume that data scatter derives solely from the measurement process; weighting data points according to more-or-less accurately estimated measurement uncertainties. This is appropriate where the assigned measurement uncertainties account for the observed scatter in the data with reasonable probability. Examples of algorithms employing this assumption are the classical statistics algorithm of York et al. (2004) and the robust statistics 'spine' algorithm of Powell et al. (2020). The main difference between these is that the spine algorithm protects against small deviations from the model assumptions (e.g. if the analytical uncertainties are slightly mis-specified or there is some relatively minor non-analytical component of scatter).

Other linear regression approaches do not require the expected scale of data scatter to be specified in advance, but rather infer this from the dataset itself. For example, typical implementations of ordinary least-squares (OLS) regression follow this approach. This is arguably more appropriate in cases where the measurement uncertainties do not account for much of the observed scatter in the data (e.g. Ludwig, 2003). Algorithms of this type that are suitable for isochron fitting include the Model 2 regression implemented in Isoplot (Ludwig, 2000) and the Robust Model 2 algorithm of Pollard et al. (2023). This is the sense in which we mean 'the data itself' (although, admittedly, this should probably have read 'the data themselves').

We have amended this sentence in an effort to improve clarity.

Line 362: Declare the value of $^{234}$U/$^{238}$U that you are using to standardise data in the text. It is mentioned in Fig 6. You might also refer to the UID associated with Li and Tissot.

In an earlier version of the manuscript, $\delta^{238}$U was defined in a footnote as a deviation in the $^{238}$U/$^{235}$U ratio of a sample relative to the CRM-112A reference material (or equivalently CRM-145). This seems to be the most common notation used in the literature. It seems that this footnote was accidentally deleted at some point. We apologise for this and have added it

back in.

We also amend the text to state that these compiled $^{238}$U/$^{235}$U data were taken from the U isotope database (UID) of Li and Tissot (2023).

**References**

Bajo, P., Drysdale, R. N., Woodhead, J. D., Hellstrom, J. C., Hodell, D., Ferretti, P., Voelker, A. H. L., Zanchetta, G., Rodrigues, T., Wolff, E., Tyler, J., Frisia, S., Spötl, C., & Fallick, A. E. (2020). Persistent influence of obliquity on ice age terminations since the Middle Pleistocene transition. *Science*, *367*(6483), 1235–1239. https://doi.org/10.1126/science.aaw1114

Cheng, H., Adkins, J., Edwards, R. L., & Boyle, E. A. (2000). U-Th dating of deep-sea corals. *Geochimica et Cosmochimica Acta*, *64*(14), 2401–2416. https://doi.org/10.1016/S0016-7037(99)00422-6

Cheng, H., Lawrence Edwards, R., Shen, C.-C., Polyak, V. J., Asmerom, Y., Woodhead, J., Hellstrom, J., Wang, Y., Kong, X., Spötl, C., Wang, X., & Calvin Alexander, E. (2013). Improvements in $^{230}$Th dating, $^{230}$Th and $^{234}$U half-life values, and U--Th isotopic measurements by multi-collector inductively coupled plasma mass spectrometry. *Earth and Planetary Science Letters*, *371–372*, 82–91. https://doi.org/10.1016/j.epsl.2013.04.006

Cliff, R. A., Spötl, C., & Mangini, A. (2010). U--Pb dating of speleothems from Spannagel Cave, Austrian Alps: A high resolution comparison with U-series ages. *Quaternary Geochronology*, *5*(4), 452–458. https://doi.org/10.1016/j.quageo.2009.12.002

Hellstrom, J. (2006). U–Th dating of speleothems with high initial 230Th using stratigraphical constraint. *Quaternary Geochronology*, *1*(4), 289–295. https://doi.org/10.1016/j.quageo.2007.01.004

Li, H., & Tissot, F. L. H. (2023). UID: The uranium isotope database. *Chemical Geology*, *618*, 121221. https://doi.org/10.1016/j.chemgeo.2022.121221

Ludwig, K. R. (2000). User's manual for Isoplot/Ex v. 2.2. *A Geochronological Toolkit for Microsoft Excel. BGC Special Publication 1a, Berkeley*, *55*.

Ludwig, K. R. (2003). Mathematical–Statistical treatment of data and errors for $^{230}$Th/U geochronology. In B. Bourdon, S. Turner, G. M. Henderson, & C. C. Lundstrom (Eds.), *Uranium-series geochemistry* (Vol. 52, pp. 631–656). Mineralogical Society of America. https://pubs.geoscienceworld.org/msa/rimg/article-abstract/52/1/631/87473

Pollard, T., Woodhead, J., Hellstrom, J., Engel, J., Powell, R., & Drysdale, R. (2023). DQPB: software for calculating disequilibrium U–Pb ages. *Geochronology*, *5*(1), 181–196. https://doi.org/10.5194/gchron-5-181-2023

Powell, R., Green, E. C. R., Marillo Sialer, E., & Woodhead, J. (2020). Robust isochron calculation. *Geochronology*, *2*(2), 325–342. https://doi.org/10.5194/gchron-2-325-2020

Tzedakis, P. C., Drysdale, R. N., Margari, V., Skinner, L. C., Menviel, L., Rhodes, R. H., Taschetto, A. S., Hodell, D. A., Crowhurst, S. J., Hellstrom, J. C., Fallick, A. E., Grimalt, J. O., McManus, J. F., Martrat, B., Mokeddem, Z., Parrenin, F., Regattieri, E., Roe, K., & Zanchetta, G. (2018). Enhanced climate instability in the North Atlantic and southern Europe during the Last Interglacial. *Nature Communications*, *9*(1), 1383–14. https://doi.org/10.1038/s41467-018-06683-3

Wieser, M. E., & Schwieters, J. B. (2005). The development of multiple collector mass spectrometry for isotope ratio measurements. *International Journal of Mass Spectrometry*, *242*(2), 97–115. https://doi.org/10.1016/j.ijms.2004.11.029

Woodhead, J., Hellstrom, J., Maas, R., Drysdale, R., Zanchetta, G., Devine, P., & Taylor, E. (2006). U–Pb geochronology of speleothems by MC-ICPMS. *Quaternary Geochronology*, *1*(3), 208–221. https://doi.org/10.1016/j.quageo.2006.08.002

York, D., Evensen, N. M., Martínez, M. L., & De Basabe Delgado, J. (2004). Unified equations for the slope, intercept, and standard errors of the best straight line. *American Journal of Physics*, *72*(3), 367–375. https://doi.org/10.1119/1.1632486

---

## Author Response (AR1)

**Author's response: 'Double dating in the Middle Pleistocene: assessing the consistency and performance of the carbonate U–Th and U–Pb dating methods'**

Timothy Pollard, on behalf of all co-authors

April 17, 2025

We thank Norbert Frank for his careful reading of the manuscript and encouraging comments. We are pleased to submit a revised version of the manuscript that addresses the Reviewers' comments as discussed in our separate responses to them.

The main changes in the revised manuscript may be summarised as follows:

- We have removed the term 'double dating' from the title.
- We have re-written the abstract to make it more streamlined and precise.
- We have re-written parts of the introduction in order to improve its logical progression and provide better context for our study. We also now explicitly acknowledge that both dating methods are based on the initial part of the $^{238}$U decay series, and that this limits our ability to verify their absolute accuracy in this study. We have also provided more background on our evaluation of speleothem $^{238}$U/$^{235}$U values.
- We have added an extra sentence on our decision not to apply a correction for initial $^{230}$Th (section 3.3).
- We have included more details on our age consistency calculations (section 4.1). This includes adding full details on age uncertainty and uncertainty correlation calculations (Appendix) for completeness.
- For consistency we now used the algebraic uncertainty propagation approach for calculating all ± 2 SE age uncertainties presented in Figure 1 and Tables 1 and 2. We also used these updated values in the collective assessment of age consistency via linear regression (section 4.1) and our assessment of average U–Pb age precision (section 4.3). This results in very minor changes to some values compared to those given in the original submission. This has no effect on any of the findings of the study.
- We have included a short section (4.4) discussing inherited $^{207}$Pb/$^{206}$Pb variability in our samples.
- We have re-written the conclusion using dot points to make it clearer and avoid excessive repetition from the abstract/introduction.

We hope these changes have significantly improved readability of the manuscript.